# Complement receptor CD46 co-stimulates optimal human CD8+ T cell effector function via fatty acid metabolism

Giuseppina Arbore[1,2], Erin E. West[3], Jubayer Rahman[3], Gaelle Le Friec[2], Nathalie Niyonzima[3,4], Mehdi Pirooznia [3], Ilker Tunc[3], Polychronis Pavlidis[2], Nicholas Powell[2], Yuesheng Li[3], Poching Liu[3], Aude Servais[5], Lionel Couzi[6], Veronique Fremeaux-Bacchi[7], Leo Placais[3], Alastair Ferraro[8], Patrick R. Walsh[9], David Kavanagh[9], Behdad Afzali [3,10], Paul Lavender[2], Helen J. Lachmann[11] & Claudia Kemper[2,3,12]

The induction of human CD4+ Th1 cells requires autocrine stimulation of the complement receptor CD46 in direct crosstalk with a CD4+ T cell-intrinsic NLRP3 inflammasome. However, it is unclear whether human cytotoxic CD8+ T cell (CTL) responses also rely on an intrinsic complement-inflammasome axis. Here we show, using CTLs from patients with CD46 deficiency or with constitutively-active NLRP3, that CD46 delivers co-stimulatory signals for optimal CTL activity by augmenting nutrient-influx and fatty acid synthesis. Surprisingly, although CTLs express NLRP3, a canonical NLRP3 inflammasome is not required for normal human CTL activity, as CTLs from patients with hyperactive NLRP3 activity function normally. These findings establish autocrine complement and CD46 activity as integral components of normal human CTL biology, and, since CD46 is only present in humans, emphasize the divergent roles of innate immune sensors between mice and men.

[1] Division of Immunology, Transplantation and Infectious Diseases, San Raffaele Scientific Institute, Milano, Italy. [2] School of Immunology and Microbial Sciences, King's College London, London, UK. [3] Laboratory of Molecular Immunology and the Immunology Center, National Heart, Lung, and Blood Institute (NHLBI), National Institutes of Health (NIH), Bethesda, MD, USA. [4] Department of Clinical and Molecular Medicine, Norwegian University of Science and Technology, Trondheim, Norway. [5] Service de Néphrologie adulte, Hôpital Necker, Paris, France. [6] Nephrologie,Transplantation, Dialyse, CHU Bordeaux, and CNRS-UMR 5164 Immuno ConcEpT, Université de Bordeaux, Bordeaux, France. [7] Assistance Publique-Hôpitaux de Paris, Hôpital Européen Georges Pompidou, and INSERM UMR S1138, Centre de Recherche des Cordeliers, Paris, France. [8] Department of Renal Medicine, Nottingham University Hospitals, NHS Trust, Nottingham, UK. [9] National Renal Complement Therapeutics Centre, Institute of Cellular Medicine, Newcastle University, Newcastle upon Tyne, UK. [10] Immunoregulation Section, Kidney Disease Branch, National Institute of Diabetes and Digestive and Kidney Diseases (NIDDK), NIH, Bethesda, MD, USA. [11] UK National Amyloidosis Centre, Division of Medicine, University College London, Royal Free Campus, London, UK. [12] Institute for Systemic Inflammation Research, University of Lübeck, Lübeck, Germany. These authors contributed equally: Giuseppina Arbore, Erin E. West, Jubayer Rahman Correspondence and requests for materials should be addressed to H.J.L. (email: H.lachmann@ucl.ac.uk) or to C.K. (email: Claudia.kemper@nih.gov)

The liver-derived serum-circulating complement system is classically recognized as a key sensor system that is required for the rapid detection and removal of pathogenic microbes. Activation of C3 into C3a and C3b and of C5 into C5a and C5b upon microbe sensing mediates the opsonization and subsequent uptake of the pathogen by scavenger cells, the migration and activation of innate immune cells to the pathogen entry side, and the initiation of the general inflammatory response[1]. Importantly, the complement system serves an equally profound role in the direct regulation of human CD4$^+$ T-cell responses. Optimal T helper type 1 (Th1) induction from CD4$^+$ T cells in rodents and humans requires T-cell receptor (TCR) activation, co-stimulatory signals and environmental interleukin (IL)-12[2]. However, while CD28-ligation signals provide largely sufficient T-cell co-stimulation in mouse T cells[3], additional signals delivered by the complement regulators/receptor CD46 (membrane cofactor protein, MCP) and the C3a receptor (C3aR) are essential to normal Th1 induction in humans[4–6].

Unexpectedly, many T-cell-modulating functions of complement are independent of serum-circulating complement and are instead driven by T-cell-generated, autocrine, complement activation fragments, which engage complement receptors expressed within the cell's interior compartments and on the surface of T cells (Supplementary Fig. 1a). Specifically, during TCR activation, C3 is cleaved intracellularly by the protease cathepsin L, which leads to intracellular as well as surface secreted C3a and C3b generation[7]. C3a binds to the G protein-coupled receptor (GPCR) C3aR and C3b engages the complement receptor and regulator CD46[8]. These receptors can be expressed intracellularly and extracellularly by the T cells and are engaged during T-cell activation in an autocrine manner. CD46 is a signaling transmembrane protein and expressed as discrete isoforms bearing one of two distinct cytoplasmic domains, CYT-1 or CYT-2—with CYT-1 driving interferon (IFN)-γ induction in CD4$^+$ T cells[9]. Autocrine CD46 engagement during T-cell stimulation drives nutrient influx and assembly of the lysosomal mammalian target of rapamycin complex 1 (mTORC1) and the glycolytic and oxidative phosphorylation metabolic pathways specifically required for IFN-γ secretion and Th1 lineage induction (Supplementary Fig. 1a)[9,10].

CD46 engagement simultaneously also triggers intracellular C5 cleavage into C5a and C5b within CD4$^+$ T cells. The stimulation of intracellular C5aR1 receptor by C5a drives the generation of reactive oxygen species (ROS) and via this the assembly of the canonical NLR family pyrin domain containing 3 (NLRP3) inflammasome in CD4$^+$ T cells[11]. NLRP3 inflammasome-generated mature IL-1β further supports IFN-γ generation in CD4$^+$ T cells and sustains Th1 responses in tissues in an autocrine fashion (Supplementary Fig. 1a). Accordingly, C3- and CD46-deficient patients have severely reduced Th1 responses and suffer from recurrent bacterial and viral infections[12,13], while mice lacking NLRP3 expression in CD4$^+$ T cells have diminished Th1 activity during lymphocytic choriomeningitis virus infection[11]. Uncontrolled intracellular C3 activation in CD4$^+$ T cells, on the other hand, has been shown to contribute to the pathologically increased Th1 responses that accompany several autoimmune diseases[4,14]. Importantly, these C3-driven responses can be pharmacologically normalized with a cell-permeable cathepsin L inhibitor that reduces intracellular C3 activation back to normal levels[7]. Aligning with a key role for the NLRP3 inflammasome in sustaining the human Th1 response, CD4$^+$ T cells from patients that suffer from cryopyrin-associated periodic syndromes (CAPS) due to gain-of-function mutations in *NLRP3*, display also significantly increased IFN-γ production. The hyperactive Th1 immunity in CAPS patients can be corrected (at least in vitro) through inhibition of the T-cell-intrinsic NLRP3 inflammasome with the specific pharmacological NLRP3 inhibitor MCC950[11,15].

Thus, an intracellular complement (the complosome) and NLRP3 inflammasome crosstalk operating at minimum through the regulation of key metabolic pathways is required for normal human Th1 induction in CD4$^+$ T cells (Supplementary Fig. 1a). Importantly, CD46 is not expressed on somatic tissues in rodents and a functional homolog has not yet been identified[8]. In addition, NLRP3 expression in mouse CD4$^+$ T cells seems to be required for Th2 responses[16], whilst there is currently no evidence that NLRP3 supports human Th2 activity[11]. Altogether these observations suggest that the (innate) pathways controlling CD4$^+$ T-cell responses are profoundly different between mice and men and that a better understanding of the human-specific drivers will likely inform on the divergent molecular mechanism underlying human T-cell-driven pathological conditions and should lead to improved (preclinical) animal models for the development of better therapeutic interventions targeting such diseases.

While the instructive role of complement in human CD4$^+$ T-cell immunity is now being increasingly recognized, there is little knowledge about complement's impact on human CD8$^+$ T-cell responses[17,18]. Given the critical roles of CD46 and NLRP3 in IFN-γ production by CD4$^+$ T cells, here we assess whether an intrinsic complosome-NLRP3 inflammasome axis is also required for the induction of cytokine production and/or cytotoxicity in CD8$^+$ T cells. Surprisingly, and opposed to CD4$^+$ T cells, CD46 stimulation is not obligatory for the initial induction of IFN-γ and tumor necrosis factor (TNF)-α production and cytotoxicity in CD8$^+$ T cells. However, CD46 is critical for the optimal induction of CTL activity (IFN-γ secretion and cytotoxicity) and emerges as superior co-stimulator that is required for normal levels of amino acid (AA) influx and subsequent mTOR activation as well as augmentation of fatty acid synthase (FASN) expression during TCR activation. Furthermore, although CD8$^+$ T cells indeed express NLRP3, neither inhibition nor potentiation of NLRP3 inflammasome function alters CD8$^+$ T-cell activity. Thus, autocrine complement and the inflammasomes play important but divergent roles in human CD4$^+$ and CD8$^+$ T cells. These findings help explaining why CD46-deficient patients suffer from recurrent viral infections[11,13] and could also indicate that NLRP3 inflammasome-targeted therapeutics may not directly interfere with protective CTL activity.

## Results

**CD46 provides superior support for CTL effector function.** We have previously shown that autocrine activation of CD46 in human CD4$^+$ T cells during TCR stimulation drives metabolic reprogramming and the assembly of an intrinsic NLRP3 inflammasome, both events critically required for normal Th1 induction and contraction[11] (Supplementary Fig. 1a). We hypothesized the existence of a functionally important similar system in human CD8$^+$ T cells. In freshly isolated healthy donor CD8$^+$ T cells (CD8$^+$/CD56$^-$) we first observed that surface CD46 is expressed at levels comparable to those of CD4$^+$ T cells (Fig. 1a), with a similar isoform pattern with regards to the cytoplasmic tail expression (CYT-1 versus CYT-2) (Supplementary Fig. 1b). Functionally, engagement of CD46 by antibodies during T-cell activation significantly enhanced production of the key effector cytokines IFN-γ and TNF-α in both CD8$^+$ and CD4$^+$ T cells when compared to CD3 or CD3 + CD28 stimulation (Fig. 1b). In addition, CD46 stimulation significantly increased degranulation, as measured by appearance of CD107a on the cell surface, granzyme B expression and killing capacity in CD8$^+$ T cells (Fig. 1c–e and Supplementary Fig. 1c). Of note, CD46 activation without concurrent CD3 stimulation had no measurable impact on CD8$^+$ T-cell activation, suggesting that CD46 fulfills an

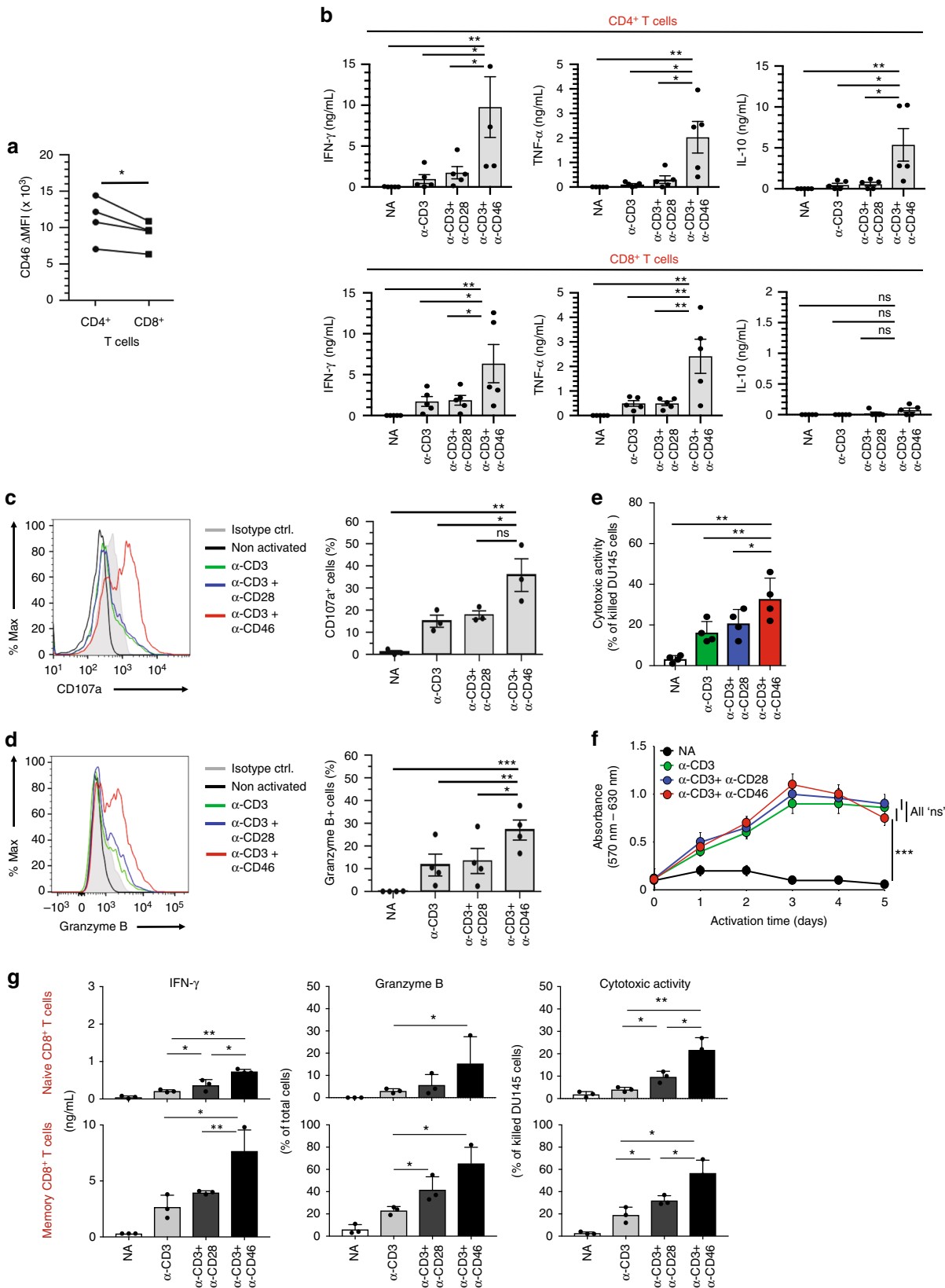

important co-stimulatory role in CTLs. Importantly, while CD46 is also critical for the co-induction of immunosuppressive IL-10 in human Th1 cells as part of Th1 contraction program[4], CD46 engagement on CD8[+] T cells failed to induce IL-10 (Fig. 1b). Likewise, CD46 stimulation, did not augment CD8[+] T-cell

proliferation and had no effect on cell viability when compared to CD3 or CD3 + CD28 activation (Fig. 1f and Supplementary Fig. 1d). Finally, CD46 significantly augmented the amount of IFN-γ secretion and numbers of CD107a and granzyme B double-positive cells, as well as cytotoxic activity when compared

**Fig. 1** CD46 co-stimulation provides superior support for CTL activity. **a** CD46 expression on the surface of resting human CD4[+] and CD8[+] T cells assessed by FACS analysis ($n = 4$, gating strategy in Supplementary Fig. 7a). **b** Comparison of IFN-γ, TNF-α and IL-10 secretion by CD3 + CD46-activated T cells. Purified CD4[+] and CD8[+] T cells from healthy donors were left non-activated (NA) or stimulated with immobilized antibodies to CD3, CD3 + CD28 or CD3 + CD46 and cytokines measured 60 h post activation ($n = 5$). **c, d** Degranulation (CD107a staining, (**c**)) and granzyme B expression (**d**) by CD8[+] T cells upon CD46 co-stimulation. CD8[+] T cells were stimulated as in (**a**) and CD107a and granzyme B expression assessed with left panels showing representative cytometry images and right panels corresponding quantifications ($n = 3$, gating strategy in Supplementary Fig. 7b). **e** Killing activity of CD46-activated CD8[+] T cells. T cells were stimulated as depicted for 24 h and cytotoxic activity of differently activated CD8[+] T cells towards DU145 target cells assessed 24 h post co-culture of T cells and DU145 cells ($n = 4$, gating strategy in Supplementary Fig. 7c). **f** Effect of CD46 co-stimulation on CD8[+] T-cell proliferation. Purified T cells were activated as indicated for 5 days and cell proliferation measured each day ($n = 4$) (black circles, non-activated cells; green, blue, and red circles, CD3, CD3 + CD28 or CD3 + CD46-activated cells, respectively). **g** Effects of CD46 co-stimulation on naive and memory CTLs. Purified human naive or memory CD8[+] T cells (sorting gates in Supplementary Fig. 7d) were activated under depicted conditions for 60 h and IFN-γ secretion, CD107a and granzyme B surface expression, and cytotoxic activity measured ($n = 3$). Error bars denote mean ± SEM (standard error of the mean). *$p < 0.05$; **$p < 0.01$; ***$p < 0.005$; ns, statistically not significant. Statistical analyses were performed using One-way ANOVA with Tukey Multiple Comparison test

to CD28 co-stimulation in both sorted naive and memory CD8[+] T cells (Fig. 1g).

To assess if CD46 engagement provides more potent co-stimulation regardless of TCR and/or CD28 signal strength, we titered the amounts of antibodies used to activate CTLs and measured subsequent IFN-γ secretion. These experiments demonstrated that amplified TCR and CD28 stimulation proportionally increased cytokine production by CTLs but that CD46 remained the strongest stimulus throughout the activation conditions tested (Fig. 2a). Moreover, we had previously shown that autocrine CD46 activation on CD4[+] T cells by TCR and CD28-driven generation of C3b is an integral and non-redundant part of human Th1 cell induction[9]. Such autocrine CD46 activation machinery seems to also exist in CD8[+] T cells: CD8[+] T cells contained intracellular stores of the C3aR, C3, and C3a as previously observed in CD4[+] T cells (Fig. 2b–c and Supplementary Fig. 2a)[7]. Also in line with our observations in CD4[+] T cells[7], activation of CTLs with increasing amounts of anti-CD3 and/or anti-CD28 antibodies was associated with greater surface deposition of C3b as well as a simultaneous proportional increase in CD107a and granzyme B expression (Fig. 2d) and IFN-γ secretion (Fig. 2e). In addition, activation of CTLs in the presence of a cell-permeable cathepsin L inhibitor (which prevents intracellular C3 activation in CD4[+] T cells[7]) also reduced C3b surface levels, degranulation and granzyme B expression, and IFN-γ secretion significantly in CTLs (Supplementary Fig. 2b, c). This strongly suggests that modulation of CTL activity by varying TCR and co-stimulatory signal strength was indeed impacted by autocrine CD46-mediated signaling. It should be noted though that the effects of cathepsin L inhibition on diminished CTL activity may not solely be attributable to reduction in intracellular C3 activation as we cannot exclude that cathepsin L inhibition had also a direct effect—at minimum—on perforin activation and hence cytotoxic activity in CTLs[19].

Collectively, these data indicate that autocrine CD46 ligation, driven by TCR and CD28-mediated generation of C3b, provides superior co-stimulatory signals that support effector functions in human CD8[+] T cells.

**CD46 signals synergize with TCR signals for CTL activity**. The complete absence of autocrine CD46 engagement in CD4[+] T cells impedes human Th1 induction even in the presence of normal TCR and CD28 signals[9,13]. To define whether CD46 activity is also obligatory for the induction of effector function in CD8[+] T cells, we utilized T cells from four CD46-deficient patients (as CD46 inhibitors or blocking reagents are not available) (Supplementary Table 1). These patients suffer from atypical hemolytic uremic syndrome and impaired Th1 responses[13,20–22]. All

patients show no or barely detectable CD46 expression on CD8[+] T cells but normal CD3 and CD28 expression levels (Fig. 3a, Supplementary Fig. 3a and Supplementary Table 2). Also, their respective proportion of memory CD8[+] T cells falls within the typical range when compared to age- and sex-matched healthy donors (Supplementary Fig. 3b), indicating that CD46 deficiency is not accompanied by a developmental CD8[+] T-cell defect. Figure 3a–d shows the analysis of all CD4[+] and CD8[+] T-cell effector functions in detail for Patient CD46-1. As previously demonstrated, CD4[+] T cells from this patient produced negligible IFN-γ (or IL-10)[13] under any activation conditions tested when compared to a healthy donor (Fig. 3b). Unexpectedly though, CD3 stimulated CD8[+] T cells from this patient demonstrated unimpaired IFN-γ secretion (Fig. 3b). TCR triggering in the presence of either CD28 or CD46 stimulating antibodies, however, failed to induce the optimal IFN-γ secretion that is seen in CTLs from healthy donors (Fig. 3b). The CD3 and CD3 + CD28-triggered basal levels of degranulation, granzyme B expression and cytotoxicity were unaltered in the patient's CD8[+] T cells but no additional increment was observed with CD46 co-ligation (Fig. 3c, d). Importantly, analyses of CTL responses from three additional CD46-deficient patients (CD46-2 to -4[13,21,22]) corroborated these observations and confirmed a significant reduction in IFN-γ production upon CD28 or CD46 co-stimulation (Fig. 3e) and lack of CD46-driven increase in degranulation and granzyme B expression (Fig. 3f and Supplementary Fig. 3c) by CD8[+] T cells with inadequate CD46 expression. CTL responses from the patients remained "depressed" also under stronger TCR and CD28 co-stimulation conditions. Due to limiting cell numbers obtained from these rare patients, we were only able to perform direct killing assays with CTLs from patients CD46-1 and -2 (Supplementary Fig. 3d). However, given that the level of degranulation (as measured by CD107a expression) correlated significantly with cytotoxic activity in CTLs (Supplementary Fig. 3e), we suggest that it is appropriate to extrapolate that optimal killing potency was also severely reduced in the CTLs isolated from patients' CD46-3 and -4 due to absent autocrine CD46 activation.

Silencing of CD46 expression in CD8[+] T cells isolated from healthy donors using siRNA (Supplementary Fig. 3f, general CD46 knockdown efficiency was 40 ± 8%) further supported the data obtained using cells from CD46-deficient patients as it also led to sub-optimal IFN-γ secretion, degranulation and granzyme B expression (Fig. 3g–h) without affecting cell viability (Supplementary Fig. 3g).

Altogether, these data demonstrate distinct functions for CD46 in CD4[+] and CD8[+] T cells: CD46 is obligatory for Th1 effector induction but not for basal CTL activity, where it functions instead to optimize effector responses.

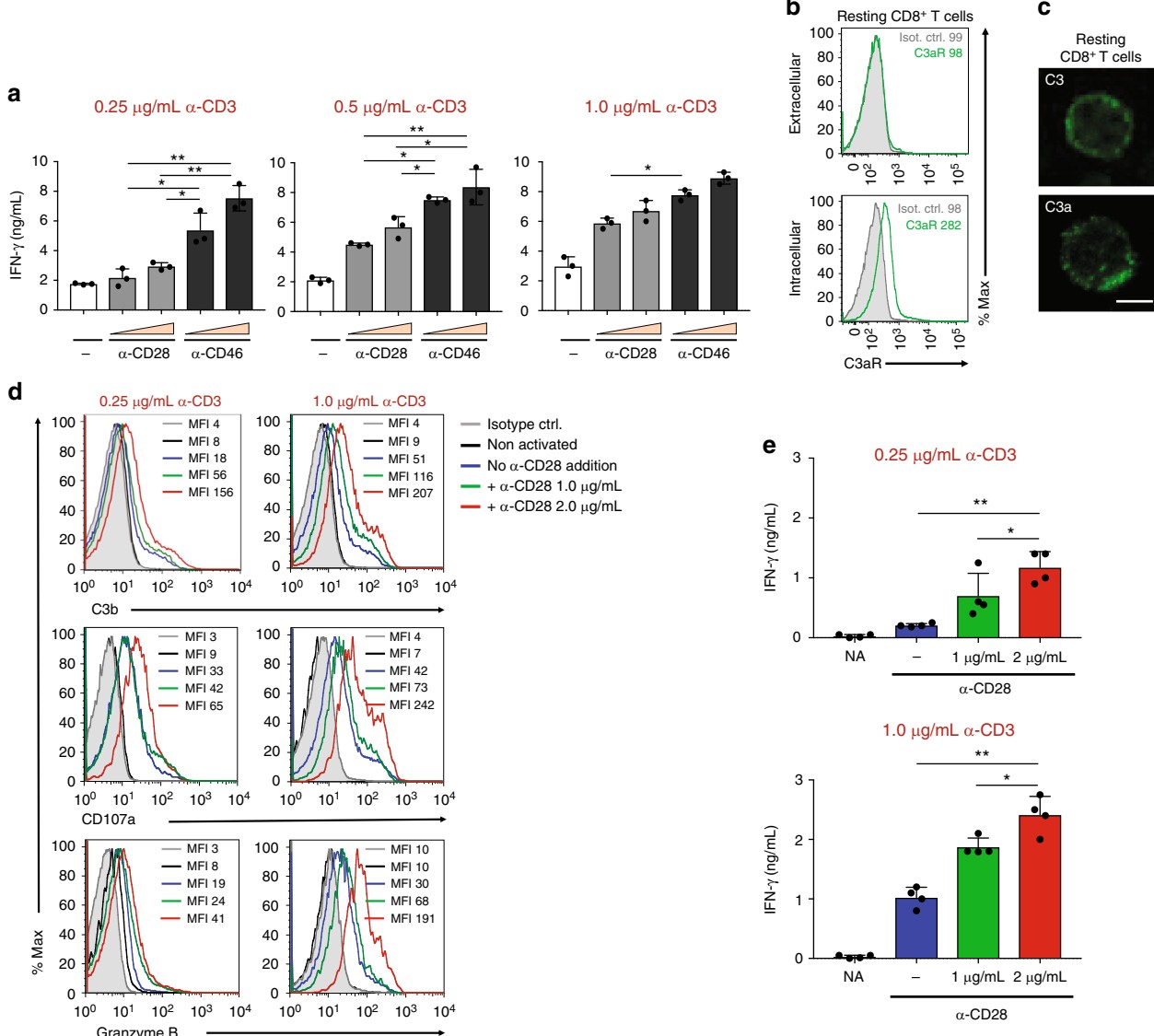

**Fig. 2** Autocrine CD46 engagement supports CTL activation. **a** Comparison of effects of TCR stimulation as well as CD28 and CD46 co-stimulation on IFN-γ secretion by CTLs. Purified CD8$^+$ T cells were activated with the depicted antibody combinations (for CD28 and CD46 either 1 μg/mL or 2 μg/mL at all tested CD3 stimulation conditions) for 60 h and IFN-γ secretion measured ($n = 3$). **b, c** Extracellular and intracellular C3aR, C3, and C3a presence in resting human CD8$^+$ T cells. Freshly purified CD8$^+$ T cells were either left non-permeabilized or were permeabilized and stained with antibodies to the C3aR, to C3 and C3a and evaluated by FACS (**b**) or confocal microscopy (**c**, scale bar = 7 μm) for presence and localization of the respective antigens (data shown are one representative of $n = 3$). **d, e** Effects of increasing TCR and CD28 activation on C3b generation, CD107a and granzyme B expression and cytokine secretion. CD8$^+$ T cells were left non-activated (NA) or activated with the depicted antibody combinations and C3b surface presence, CD107a and granzyme B positivity (**d**, shows a representative FACS analysis of two similarly performed experiments; gray histograms, isotype controls; black histograms, non-activated cells; blue histograms, no CD28 stimulation; green and red histograms, addition of 1 or 2 μg/mL α-CD28, respectively) and IFN-γ production (**e**) measured at 12 h post activation ($n = 3$). The gating strategy for CD8$^+$ T cells flow cytometry staining is shown in Supplementary Fig. 7b. Error bars denote mean ± SEM. *$p < 0.05$; **$p < 0.01$; ns, statistically not significant; ctrl. Control. Statistical analyses were performed using One-way ANOVA with Tukey Multiple Comparison test

**NLRP3 function is distinct in CD4$^+$ and CD8$^+$ T cells**. We next assessed the mechanisms by which CD46 may augment CTL function. In CD4$^+$ T cells, intrinsic NLRP3 inflammasome assembly driven by intracellular C5 activation works synergistically with CD46 and is required for maintaining optimal levels of IFN-γ secretion in Th1 cells[11]– we hence analyzed CD8$^+$ T cells for the presence of this crosstalk. Indeed, CD8$^+$ T cells expressed both C5a receptors, C5aR1 and C5aR2, at similar levels and locations (C5aR1 exclusively intracellularly and C5aR2 intra- and extracellularly) when compared to CD4$^+$ T cells and also contained intracellular stores of C5, from which they generated C5a

once stimulated (Fig. 4a–c). Similarly, CD8$^+$ T cells expressed NLRP3, whose expression was increased upon TCR activation particularly in conjunction with CD46 co-stimulation (Fig. 4d). However, and in contrast to CD4$^+$ T cells, neither *IL1B* gene transcription (Fig. 4e) nor active IL-1β secretion (Fig. 4f) were significantly augmented in CD8$^+$ T cells upon CD46 co-stimulation. In line with these findings, addition of the canonical NLRP3 inflammasome inhibitor MCC950[15] reduced IL-1β and IFN-γ production in CD4$^+$ T cells (Fig. 4g)[11] but had no effect on these cytokines in CD8$^+$ T cells (Fig. 4g and Supplementary Fig. 4a). Moreover, although CD8$^+$ T cells contained

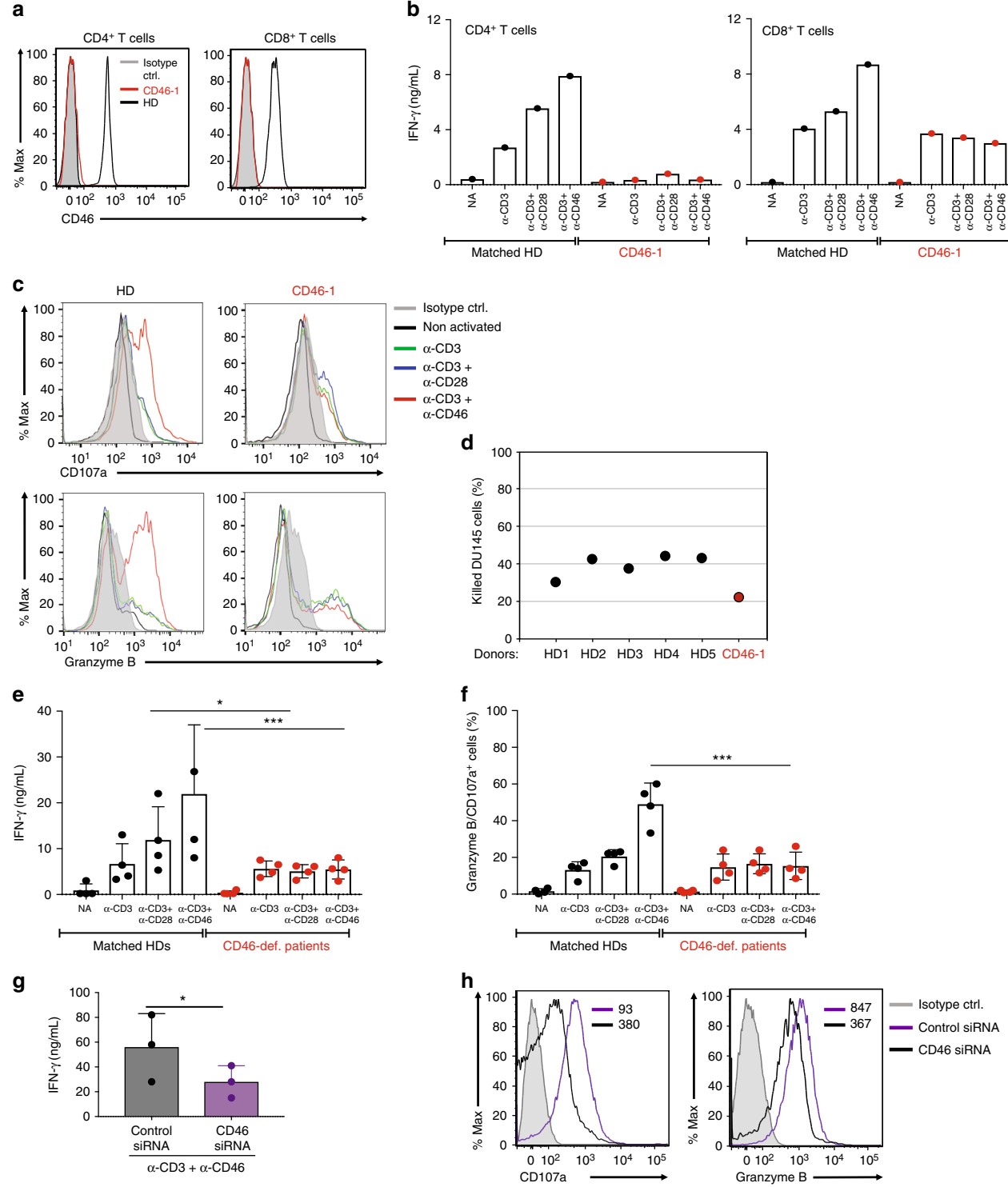

active caspase-1 (which is a key downstream product of NLRP3 inflammasome activation) with increased amounts observed under all activation conditions (Supplementary Fig. 4b), MCC950 reduced active caspase-1 generation only in CD4[+] T cells[11], but not in CD8[+] T cells (Fig. 4h). Granzyme B production also remained unaffected by MCC950 treatment (Fig. 4i) and we were unable to detect IL-18, another cytokine activated by the NLRP3 inflammasome, in CD8[+] T cells (Fig. 4j). We noticed, however, that the NLRP3 protein translocated to the nucleus in substantial amounts upon CD3 + CD46 activation (Supplementary Fig. 4c).

A divergent role for the NLRP3 inflammasome in CD4[+] and CD8[+] T cells is further supported by our finding that CD8[+] T cells from patients with CAPS, a series of autoinflammatory diseases with gain of function *NLRP3* gene mutations[23] (Supplementary Table 3), functioned normally. We had previously demonstrated that uncontrolled NLRP3 inflammasome activity in CD4[+] T cells from these patients translated into increased caspase-1 activity, IL-1β secretion and hyperactive Th1 responses, which could all be normalized with MCC950 treatment (Fig. 5a–b)[11]. In stark contrast, CD8[+] T cells from

**Fig. 3** CD46 activation has distinct outcomes in CD4+ and CD8+ T cells. **a** CD46 expression on resting CD4+ and CD8+ T cells from patient CD46-1 (red histograms) and a matched healthy donor (HD) (black histograms; gating strategy as in Supplementary Fig. 7a). **b** IFN-γ secretion by CD4+ and CD8+ T cells from patient CD46-1. Purified T cells were left non-activated (NA) or activated with the depicted immobilized antibodies and IFN-γ secretion assessed (60 h). Data shown are mean values of each condition performed in duplicate. **c** Degranulation (CD107a) and granzyme B expression of CTLs from patient CD46-1. Purified CTLs were activated with the indicated antibody combinations and CD107a and granzyme B expression measured at 60 h (gray histograms, isotype controls; black histograms, non-activated cells; green, blue, or red histograms, CD3, CD3 + CD28, or CD3 + CD46 activation, respectively; gating strategy shown in Supplementary Fig. 7b). **d** Cytotoxic activity of CTLs from patient CD46-1. CTLs from patient CD46-1 were CD3 + CD46-activated for 24 h and killing activity measured in comparison to CTLs from five HDs (gating in Supplementary Fig. 7c). **e, f** IFN-γ production and degranulation capacity of CTLs from CD46-deficient patients CD46-1 to CD46-4. Purified CD8+ T cells from patients CD46-1 to -4 and from four sex- and age-matched HDs were activated as in (**b**) and IFN-γ secretion (**e**) and CD107a and granzyme B+ cells (**f**, gating as in Supplementary Fig. 7b) measured (n = 4). **g, h** Effect of CD46 protein knockdown on HD CTL effector function. CD8+ T cells from three different healthy donors were transfected with either siRNA targeting CD46 mRNA or control siRNA for 48 h, then activated with immobilized antibodies to CD3 and CD46 for further 48 h and IFN-γ secretion (**g**) and CD107a and granzyme B expression measured (**h**, gating as in Supplementary Fig. 7b; representative cytometry plots of three similarly performed experiments; black histograms, control siRNA; purple histograms, CD46 siRNA) (n = 3). Error bars denote mean ± SEM. *p < 0.05; ***p < 0.005; ctrl.: control. Statistical analysis was performed using One-way ANOVA with Tukey Multiple Comparison test

these patients demonstrate no significant increase in IL-1β or IFN-γ secretion, degranulation, granzyme B expression or cytotoxicity (Fig. 5a–d and Supplementary Table 4). Furthermore, proportions of naive vs. memory CD8+ T cells in the patients are within the normal range (Supplementary Fig. 5a, b).

These data suggest that, although human CTLs harbor an intracellular C5 system and express NLRP3, the intracellular C5 activation-driven formation of a canonical NLRP3 inflammasome —as opposed to CD4+ T cells—is not one of the mechanisms by which CD46 signals support optimal CD8+ T-cell effector function.

**CD46 augments cell metabolism for optimal CTL activity.** To pinpoint pathways driven by CD46 co-stimulation in CTLs, we compared the gene expression profile of CD3 + CD46-activated purified CD8+ T cells from three CD46-deficient patients (CD46-2 to -4) and four sex- and age-matched healthy donors at 6 h post activation via RNA-Sequencing technique. Approximately 2300 transcripts were differentially expressed (by at least two-fold at q-value of 0.05 or less) upon CD3 + CD46 activation in CTLs from healthy donors compared to 147 transcripts in patients with CD46 deficiency (Fig. 6a and Supplementary Data 1). Nearly all of the genes altered in patient cells were also differentially expressed in healthy donors. In contrast, approximately 1900 differentially expressed transcripts were unique to healthy donors (Fig. 6b). As in CD4+ T cells[9], CD46 induced most strongly genes connected with multiple metabolic pathways in healthy donor CTLs (Supplementary Data 2). These included *FASN* (fatty-acid synthase), *FABP5* (fatty-acid binding protein 5), and *SLC7A5* (coding for the L-type large neutral amino acid transporter LAT1), which are key components of lipid/lipoprotein metabolism and membrane solute transport, respectively (marked in "red" in Fig. 6c). Comparative gene arrays using CD8+ T lymphocytes from three healthy volunteers activated in vitro with antibodies to CD3 alone versus CD3 and CD46 in combination confirmed that the identified induced pathways are mostly driven by CD46 co-stimulation and not TCR engagement alone (Supplementary Fig. 6a, b and Supplementary Data 3).

Aligning with this, CD46 significantly increased LAT1 protein expression (Fig. 6d, left panel) and activation of the amino acid sensor mammalian target of rapamycin (mTOR), measured by augmented phosphorylation of the downstream mTOR target S6 Kinase (S6K) (Fig. 6d, middle panel)[24]. mTOR-mediated activation of fatty-acid reprogramming (which involves FABP5 and FASN) is critical for CD8+ T-cell proliferation, effector function[25] and survival of tissue-resident CD8+ T cells[26,27]. Indeed, we observed that CD46 co-stimulation also increased FASN protein expression significantly when compared to CD3

and CD3 + CD28-stimulated CD8+ T cells (Fig. 6d, right panel) while reduction of CD46 expression significantly reduced *FASN* gene transcription during activation (Supplementary Fig. 6c). Furthermore, restraint of FASN activity through the specific inhibitor C75 during CD3 + CD28 and CD3 + CD46 activation reduced IFN-γ secretion and degranulation by in vitro stimulated T cells in a dose-dependent manner to about the levels observed with CD3 activation alone (Fig. 6e) without affecting cell viability (Supplementary Fig. 6d).

Importantly, equivalent induction changes in *FASN*, *FABP5*, or *SLC7A5* transcription in CD46-deficient CTLs were not evident in the RNA-Sequencing analysis (Fig. 6f) and subsequent quantitative PCR analyses showed indeed a significant diminution of FASN and FABP5 induction—and a trend for reduced SLC7A5 expression—in CD3 + CD46-stimulated CTLs from CD46-deficient patients when compared to healthy donors (Fig. 6g).

Altogether these data indicate that the mechanism by which autocrine CD46 co-stimulation enhances CTL effector function during CD8+ T-cell activation is—at minimum—through augmentations in nutrient influx, mTOR activity, and fatty-acid metabolism (Fig. 7).

## Discussion

Here, we explored the role of CD46 signaling in human CD8+ T cells. A number of studies in recent years have prompted substantial changes in our perception of the roles of complement. It is now well recognized that local complement production clearly orchestrates basic physiological processes of CD4+ T cells, including cellular metabolism, survival, proliferation, and autophagy[28,29]. Importantly, CD46 is ubiquitously expressed by human, but not murine, immune cells[30] and regulates activation and secretion of intracellular C3 and C5[7]. This self-contained CD46-regulated autocrine-functioning C3–and–C5–system (the complosome) is directly responsible for induction and contraction of human Th1 responses via direct regulation of nutrient influx, glycolysis, mitochondrial activity, and inflammasome assembly[31–33]. Patients with genetic CD46 (or C3) deficiency suffer from recurrent bacterial and viral infections[7], thus we reasoned that the complosome may also be critical for the function of human CD8+ T cells, a previously unexplored notion.

Our findings indicate the presence of a full-blown functional complosome in CD8+ T cells, including surface CD46, intracellular stores of C3 and C5 and generation of intracellular C3a and C5a fragments upon T-cell activation, comparable to human CD4+ T cells[11]. Moreover, we identified CD46 as a novel, key co-stimulatory molecule driving optimal induction of both human naive and memory CTL effector responses, including IFN-γ

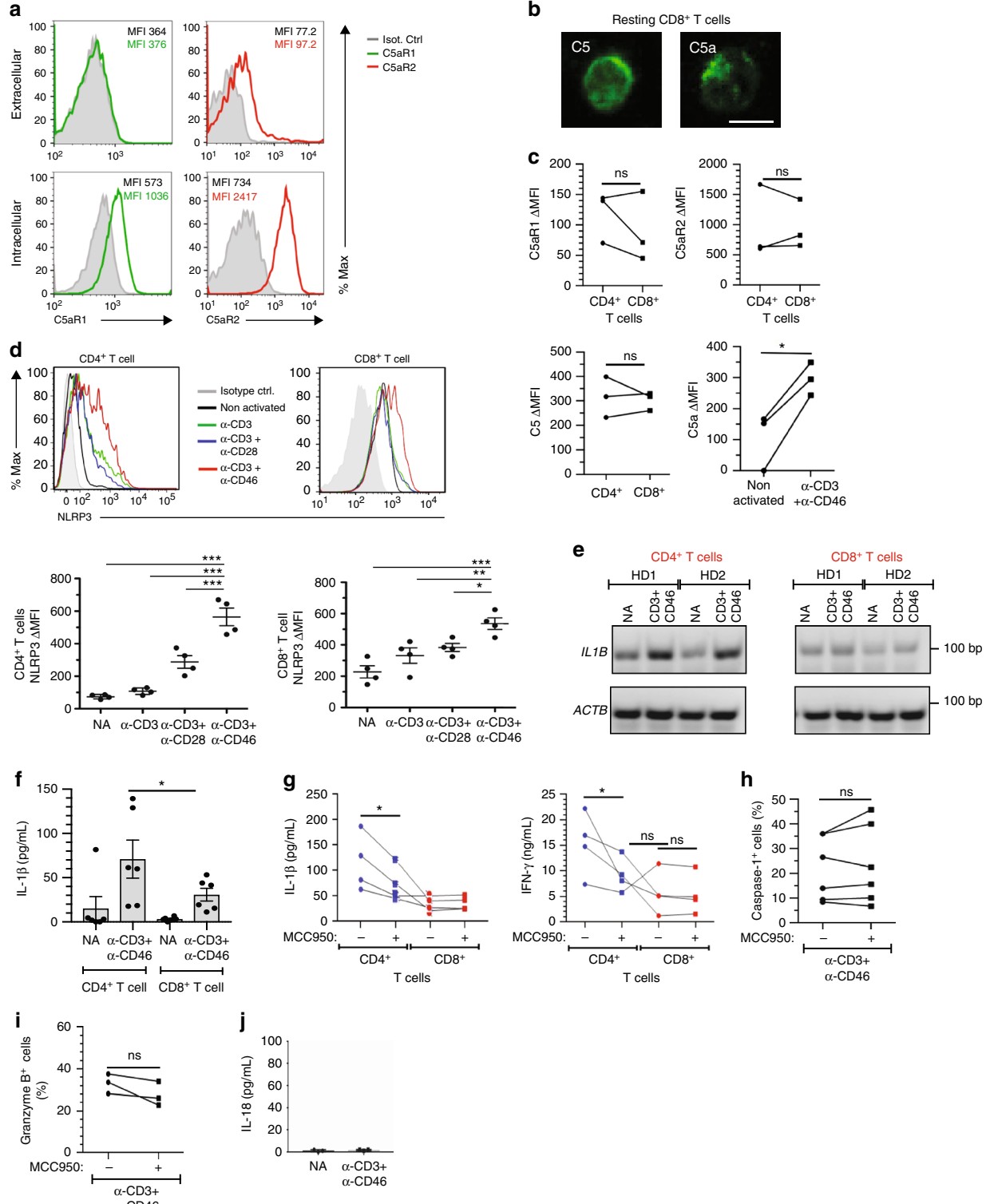

production and cytotoxic activity. Importantly, autocrine CD46 engagement on CTLs, similar to that on CD4⁺ T cells, is driven by TCR and CD28-supported intrinsic C3b generation, and is a requirement for optimal CTL responses. A C3b-interacting complement receptor/regulator that functions as murine homolog to human CD46 with regards to Th1 regulation and CTL activation has not been identified, and current studies indicate that CD28 is sufficient to drive glycolysis and OXPHOS needed for the induction of CD4⁺ T-cell effector function in mice[31]. This

further substantiates the growing understanding that the molecular mechanisms supporting successful T-cell activation downstream of key co-stimulators is distinct between mice and men.

Dysregulated CD46 engagement in CD4⁺ T cells contributes to several Th1-mediated autoimmune states[7,34] and a recent study indicates that a novel mutation in the *CD46* gene is also associated with aberrant CTL responses in multiple sclerosis[35], suggesting strongly that our findings here have indeed implication for CTL-driven diseases in humans. In mice, modulating intrinsic

**Fig. 4** Divergent roles for NLRP3 in CD4[+] and CD8[+] T cells. **a, b** Extracellular and intracellular C5aR1, C5aR2, C5 and C5a in resting human CD8[+] T cells. Freshly purified CD8[+] T cells were evaluated for extra- and intracellular C5aR1 (green histograms), C5aR2 (red histograms), C5 and C5a expression by FACS (**a**) or confocal microscopy (**b**, scale bar = 7 μm). **c** Levels of C5aR1, C5aR2, and C5 expression in resting CD4[+] and CD8[+] T cells and C5a generation in activated CD8[+] T cells (n = 3). **d** Effect of CD46 co-stimulation on NLRP3 expression in T cells. T cells from healthy donors were left non-activated (NA) or stimulated with the depicted antibodies and NLRP3 expression measured 60 h post activation (n = 4). Upper panels show a representative cytometry plot of NLRP3 expression and lower panels the corresponding quantifications (gray histograms, isotype controls; black histograms, non-activated cells; green, blue, or red histograms, CD3, CD3 + CD28, or CD3 + CD46 activation, respectively; gating strategy for Fig. 4a–d is shown in Supplementary Fig. 7b). **e, f** Effect of CD46 co-stimulation on *IL1B* gene transcription at 6 h post activation ((**e**) n = 2)) and IL-1β secretion ((**f**) n = 4)) in CD4[+] and CD8[+] T cells at 60 h post activation (for an uncropped image of panel (**e**), see Supplementary Fig. 8). **g** Impact of MCC950 on IL-1β and IFN-γ production from CD46-activated T cells. T cells were CD3 + CD46-activated with or without MCC950 (10 μM) and cytokine production determined (60 h) (n = 4). **h, i** Effects of MCC950 on active caspase-1 generation and granzyme B expression in CTLs. CTLs were treated as in (**g**) and active caspase-1 ((**h**) n = 6 and granzyme B measured (**i**, n = 3). **j** TCR and CD46 activation fail to induce IL-18 in human CTLs (72 h post activation, n = 4). Error bars denote mean ± SEM. *p < 0.05; **p < 0.01; ***p < 0.005; ns, not significant; ctrl, control. Statistical analyses were performed using One-way ANOVA with Tukey Multiple Comparison test or the Paired Student's t-test where appropriate

C3 activity (the equivalent of C3b-mediated CD46 engagement in humans) in tumor-infiltrating lymphocytes can promote tumor progression[36]. Altogether, these studies and our current work suggest that understanding human-specific complement pathways in CTL biology could open avenues to novel therapeutic approaches in cancer, vaccine development and/or autoimmunity.

Mechanistically, CD46 signaling augmented TCR-induced influx of amino acids, mTOR activation and lipid biosynthesis, requisite events for normal CTL function[27]. CD46 deficiency, either genetic or experimentally induced by protein knockdown in healthy donor T cells, caused sub-optimal lytic activity and IFN-γ production in CTLs, which would explain the predisposition of CD46-deficient patients to recurrent viral infections[7] as viral control requires both optimal CD4[+] and CD8[+] T-cell activity.

Of note, an earlier study observed CD46-driven increase in CD25, CD28, and tumor necrosis factor receptor superfamily, member 4 (TNFRSF4 or OX40, CD134) expression in CD8[+] T cells from healthy donors but no augmentation of IFN-γ responses (effects on CTL cytotoxic activity were not assessed)[17]. However, this study did not deplete CD56[+] cells (natural killer (NK) and NK T cells) from their cultures, which impact heavily on CD8[+] T-cell function, and could explain the discrepancy in observed outcomes of CD46 co-stimulation of CTLs between the two studies. This is supported by recent work published by Hansen and colleagues that confirms our finding as this group also observed increased IFN-γ secretion by human CTLs activated in the presence of anti-CD46 antibodies[37]. Cytotoxic activity, or CD46-driven downstream signals, however, were not explored in this work.

Interestingly, although CD46 was a key co-stimulator in both populations, we found that the role of CD46 differs in CTLs compared to helper T cells. In particular, CD46-mediated signals are obligatory for induction of Th1 differentiation in CD4[+] T cells[7,9,13] but not obligatory for initiation of effector function in CD8[+] T cells per se. Instead, the complosome was critical for optimal CTL activity. This may reflect inherent differences between T helper cells, which undergo fate decisions that differentiate them to specific effector lineages, and CTLs, which have more limited functional repertoires and require no additional differentiation.

Furthermore, engagement of CD46 on expanding Th1 cells induces high IL-10 co-production and is required to transition these cells into a suppressive and contracting phenotype[4]. Without this switch into IL-10 co-production, activated and expanding CD4[+] T cells fail to contract Th1 responses and remain hyperactive, as seen in rheumatoid arthritis[4], multiple

sclerosis[14], and systemic lupus erythematosus[34]. We observed that CD8[+] T cells failed to induce significant levels of IL-10 upon CD46 stimulation, possibly reflecting the short half-life of CTL responses and a dispensable role of autocrine IL-10 for contraction of these responses.

Although our current study indicates that CD46-driven signaling events are distinct between CD4[+] and CD8[+] T cells, a common theme is their impact on cell metabolic pathways regulating T-cell effector function. We have previously demonstrated that CD46 is critically required to regulate glucose and AA influx and to drive glycolysis and OXPHOS in CD4[+] T cells[9]. We demonstrate here that in CTLs, ligation of this surface protein also mediates AA influx and augments cellular fatty-acid metabolism, supporting the assertion that the complosome serves critical novel and non-canonical roles in cell biology. This notion aligns well with the growing understanding that other classical danger sensing or pattern recognition receptor (PPR) systems such as the toll-like receptors (TLRs), the nod-like receptors (NLRs) and different inflammasomes engage in the functional crosstalk with a series of metabolic checkpoint systems via inducing and/or recognizing metabolic changes to control both cell activation and homeostasis[38]. A crosstalk between the complosome and the canonical NLRP3 inflammasome that modulates oxygen metabolism is also an integral part of normal human Th1 biology: CD46 stimulation increases intracellular C5a production and intracellular C5aR1-dependent mitochondrial ROS generation, which leads to the assembly of the NLRP3 inflammasome in CD4[+] T cells[11]. NLRP3 inflammasome-induced caspase-1 activation and secretion of mature IL-1β then support sustained Th1 activity in an autocrine fashion. NLRP3 inhibition or reduction of intracellular C5aR1 expression strongly impairs human Th1 responses. Conversely, CD4[+] T cells from CAPS patients, which express constitutively-active NLRP3, produce uncontrolled IL-1β and have pathologically increased Th1 activity that can be normalized by the specific NLRP3 inhibitor MCC950[11,15]. Because CD8[+] T cells express the required complement components and receptors as well as NLRP3, our finding that an intrinsic complosome—inflammasome—axis is not required for normal IFN-γ production (or cytotoxicity) in CTLs was unexpected: although we detected active caspase-1 and low amounts of mature IL-1β in stimulated CD8[+] T cells, neither activation of caspase-1 nor production of IL-1β could be inhibited by MCC950. Furthermore, MCC950 had no effect on IFN-γ secretion or killing ability of CTLs, which aligns fully with our finding that CD8[+] T cells from CAPS patients, in contrast to their CD4[+] T cells, have no intrinsically altered phenotype. While the significance of active caspase-1 and/or low-level IL-1β secretion in human CTLs remains to be defined, TNF-α can drive NLRP3 inflammasome-

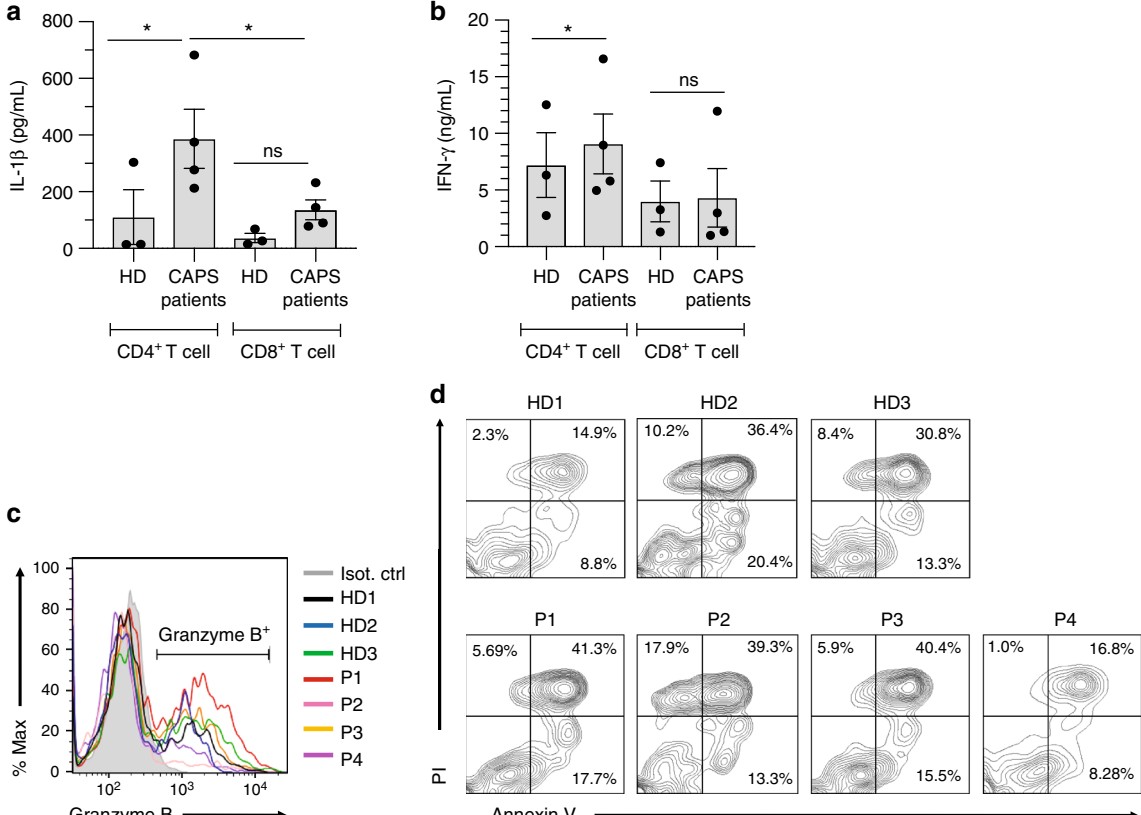

**Fig. 5** CTLs from CAPS patients function normally. **a, b** IL-1β and IFN-γ production upon CD46 co-stimulation in CD4+ and CD8+ T cells from CAPS patients. T cells from four CAPS patients and three age and sex-matched HDs were CD3 + CD46-activated and IL-1β (**a**) and IFN-γ (**b**) secretion measured at 60 h post activation. **c** Granzyme B expression in CD3 + CD46-activated CD8+ T cells from CAPS patients at 60 h post activation (gating strategy shown in Supplementary Fig. 7b; gray histogram, Isotype control; black, blue, and green histogram, HD1–3, respectively; red, pink, yellow, and purple histogram, P1-P4, respectively). **d** Cytotoxic activity of CD8+ T cells from CAPS patients. CD8+ T cells from HDs 1–3 and CAPS patients 1–4 were CD3 + CD46-activated for 24 h and their cytotoxic activity determined (gating strategy in Supplementary Fig. 7c). Error bars denote mean ± SEM. *$p < 0.05$; ns, statistically not significant. Statistical analyses were performed using One-way ANOVA with Tukey Multiple Comparison test

independent caspase-1 maturation[39] and this pathway may be operative here. The function of NLRP3 in CD8+ T cells also remains currently unresolved. We noted that substantial amounts of NLRP3 translocate to the nucleus in human CD8+ T cells upon CD3 + CD46 activation (Supplementary Fig. 4c) and future work may possibly uncover a non-canonical nuclear function for NLRP3 in human CTLs as it was previously observed in mouse CD4+ T cells[16]. Similarly, the definition of the exact role(s) of the autocrine C5 system in human cytotoxic T cells requires further work as well. Antiviral CTL activity is reduced in *C3* or *C5ar1*-deficient mice and augmented in *Daf*−/− mice (because of unrestrained local C3a and/or C5a generation as the decay acceleration factor (DAF, CD55) is a complement activation inhibitor) supporting that C3 and C5 activation fragments are indeed involved in regulating CD8+ T-cell responses[40–42]. We had previously shown that the intracellular C5 system in human CD4+ T cells induces mitochondrial ROS production, which is also required for cell proliferation[11,43], hence, assessing for a potential role of C5-regulated mitochondrial activity in CTLs may be a suitable starting point. Nevertheless, our observation here that crosstalk between the complosome and the NLRP3 inflammasome is not required for normal CTL activity in humans suggests that the NLRP3-targeted therapeutics currently under assessment will not interfere directly with protective CTL responses.

In sum, our findings establish CD46 as a key modulator of both CD4+ and CD8+ T-cell immunity through the mediation of nutrient influx and regulation of key metabolic pathways but they also further emphasize an emerging concept in the literature, namely that innate immune sensor systems have critical but divergent roles among different immune cell subpopulations and between mice and men.

## Methods

**Healthy donors and patients.** Blood samples from healthy donors and patients were obtained and processed with ethical and institutional approvals. Three patients with deficiency in CD46[13,21,22] (patients CD46-1 to -3) were recruited to King's College London (REC number 09/H0803/154; KCL Ethics Committee and Wandsworth Research Ethics Committee) and one patient (CD46-4) was recruited to the Nottingham University Hospital/National Complement Therapeutics Center (REC number 01/2/038; North East-York Research Ethics Committee) with key information on the patients summarized in Supplementary Table 1. Four adult patients with cryopyrin-associated periodic syndrome (CAPS) were recruited at the National Amyloidosis Center, University College London (UCL Ethical Approval Committee, REC reference number 06/Q0501/42) with key information on the patients summarized in Supplementary Table 3. In all experiments that involved T cells from the CD46-deficient patient or from CAPS patients, T cells from age- and sex-matched healthy volunteers were used as controls. Informed consent was obtained from all human participants of this study.

**Antibodies, agonists/antagonists and inhibitors.** Cell-stimulating monoclonal antibodies to human CD8+ T cells were bought from BD Biosciences, San Diego, CA (anti-hCD28, CD28.2, cat. 555726), purified from a specific hybridoma (anti-

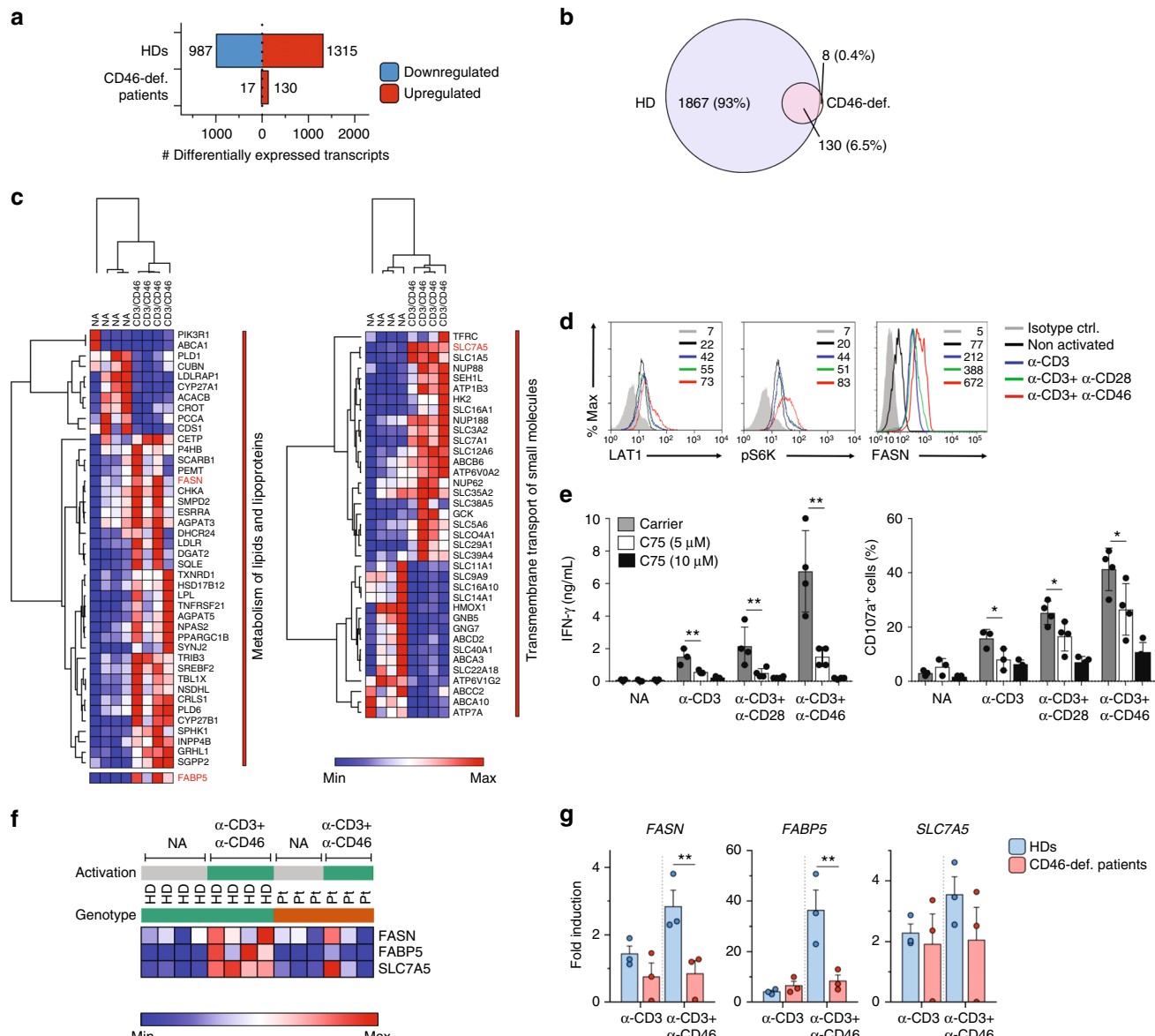

**Fig. 6** CD46 co-stimulation augments nutrient influx and fatty-acid synthesis. **a, b** Differentially expressed genes in CD8[+] T cells of four healthy donors (HD) and three CD46-deficient (CD46-def.) patients (CD46-2 to -4) upon CD3 + CD46 activation at 6 h (**a**) and Venn diagram indicating overlap between differentially expressed transcripts (**b**). **c** Heatmaps comparing non-activated (NA) and CD3 + CD46-activated (6 h) HD CTLs with focus on genes involved in lipid/lipoprotein metabolism (left), transmembrane solute transport pathways (right), and FABP5 (left below). **d** Effect of CD46 co-stimulation on LAT1 and FASN expression and on mTOR activation. CD8[+] T cells (gated as in Supplementary Fig. 7b; gray histograms, isotype controls; black histograms, non-activated cells; blue, green, or red histograms, CD3, CD3 + CD28, or CD3 + CD46 activation, respectively) from HDs were left NA or stimulated as depicted and LAT1 and FASN expression and mTOR activity (70S6K phosphorylation) measured 48 h post activation (n = 4). **e** Effect of FASN inhibition on IFN-γ production and degranulation of CTLs. CD8[+] T cells were activated as in (**d**) in the presence or absence of the FASN inhibitor C75 and IFN-γ secretion and CD107a expression measured at 48 h post activation (n = 3–4; gray bars, carrier addition; white bars, addition of 5 μM C75; black bars, addition of 10 μM C75). **f** Heatmap indicating FASN, FABP5 and SLC7A5 expression in HD (n = 4) vs. CD46-deficient (n = 3) CD8[+] T cells in resting state and after CD3 + CD46 activation. **g** RT-PCR for *FASN*, *FABP5*, and *SLC7A5* in CD3 or CD3 + CD46-activated CD8[+] T cells from patients CD46-1, CD46-2, and CD46-4 (light red bars), and three matched HDs (light blue bars) at 6 h post activation (n = 3). Error bars denote mean ± SEM. *p < 0.05; **p < 0.01. Statistical analyses were performed using One-way ANOVA with Tukey Multiple Comparison test or the Paired Student's t-test where appropriate

hCD3; OKT-3) or generated in-house (anti-CD46; TRA-2-10)[44] and used at the indicated concentrations under the T-cell stimulation protocol. Antibodies against human CD4 eFluor 450 (OKT4, cat. 48-0048-42, 0.25 μg/mL), CD46 PE (clone 8E2, cat. 12-0469-42, 2.0 μg/mL), CD56 APC (MEM-188, cat. MA1-19462, 1.0 μg/mL) and CD107a FITC (eBioH4A3, cat. 11-1079-42, 3.0 μg/mL) were bought from eBioscience (San Diego, CA). Anti-human granzyme B FITC (GB11, cat. 560211, 5.0 μL/test), C3aR PE (hC3aRZ8, cat. 561178, 3.0 μL/test), CD45RA PECy7 (5H9, cat. 561216, 2.0 μL/test), and CD45RO BV711 (UCHL1, cat. 563722, 2.0 μL/test)

antibodies were purchased from BD Biosciences (San Jose, CA, USA). Anti-human NLRP3 (ab4207, 5.0 μg/mL), C5 (ab11898, 1.0 μg/mL) and C5a neo-epitope (ab11878) were produced by Abcam (Cambridge, UK). Antibodies to human C5aR1 FITC (S5/1, Santa Cruz Biotechnology, CA, cat. sc-53795, 5.0 μg/mL), C3 (GW20073F, Sigma-Aldrich, 5.0 μg/mL), CD8a (RPA-T8, PerCPCy5.5 cat. 301032 and PECy7 cat. 301012, 2.0 μL/test) and C5aR2 PE (1D9-M12, cat. 342404, 2.0 μL/test) (both from Biolegend, Cambridge, UK), phosphorylated p70S6 Kinase (9205 S; 0.5 μg/mL) and LAT1 (5347; 1 μg/mL) (both from Cell Signaling Technology,

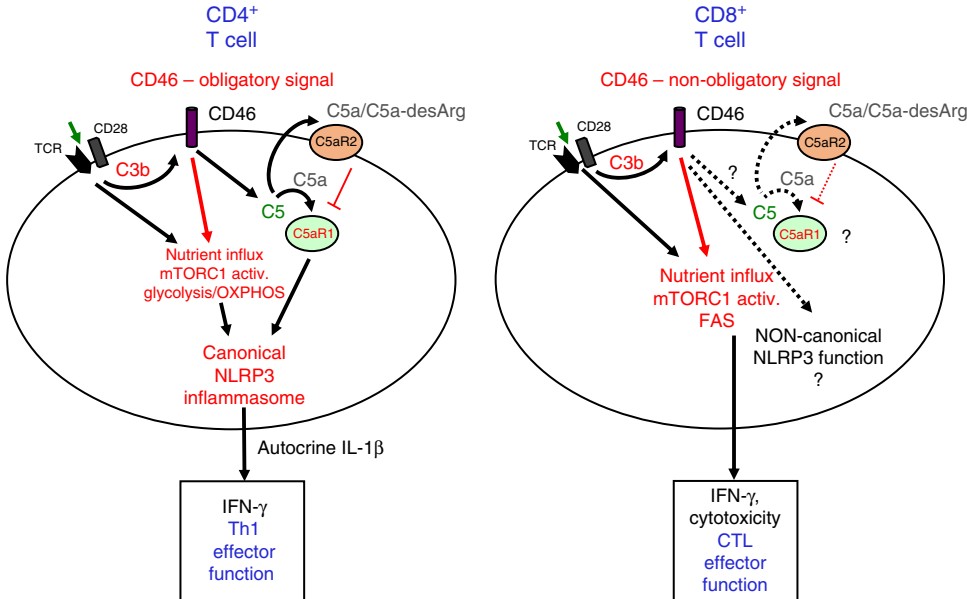

**Fig. 7** Model of complosome and inflammasome roles in CD4$^+$ and CD8$^+$ T cells. TCR and CD28 stimulation-induced C3b generation mediates autocrine CD46 stimulation in both T-cell types. This is an obligatory event for Th1 effector function induction while CD46 engagement is a superior co-stimulator on CD8$^+$ T cells. In both cases, CD46 functions through mediating nutrient influx and augmentation of key metabolic pathways. Whereas a canonical C3/C5-activated NLRP3 inflammasome in CD4$^+$ T cells sustains optimal Th1 responses (via autocrine IL-1β), such a parallel role for the canonical NLRP3 inflammasome may not exist in CTLs. Also, while surface expressed C5aR2 negatively regulates human Th1 activity, the role for the C5 system in CTLs remains to be defined (dashed arrows and questions marks). Similarly, a non-canonical functional role for NLRP3 in CTLs cannot be excluded (dashed arrows and question mark). FAS, fatty-acid synthesis

Danvers, MA) were purchased while the anti-human C3a neo-epitope antibody was a gift from Jörg Köhl (1:100 dilution; University of Lübeck, Germany)[45]. The anti-human C5a antibody was also biotinylated in house using the APEX™ Biotin-XX Antibody Labeling Kit (Life Technologies Ltd, Paisley, UK) an used at a final concentration of 0.2 μg/mL. The anti-human fatty-acid synthase (FASN) antibody (EPR7466, 0.5 μg/mL) was obtained from Abcam. The secondary antibodies used included: anti-rabbit IgG H + L chain Alexa Fluor 594 (ab150076), anti-goat IgG H + L chain PE (ab7004) and anti-goat IgG H + L chain Alexa Fluor 488 (ab150129), all from Abcam. An anti-mouse IgG Alexa Fluor 488 (A11001), and anti-chicken IgG H + L chain Alexa Fluor 488 (A11039), both from Molecular Probes (Paisley, UK) and used for the confocal microscopy experiments. The secondary antibodies were all used at 1:200 final dilution. APC-conjugated strep-tavidin was bought from Biolegend (cat. 405207) and used at a final dilution of 1.0 μg/mL.

Recombinant human IL-2 was provided by Dr. Christine Pham (Washington University in Saint Louis, MO) and the specific C5aR2 agonist (RHYPYWR) was generated by T. Woodruff and P. Monk (Sheffield University, UK)[46] and used at 100 μM. The NLRP3 inflammasome antagonist MCC950[15] was provided by M. Cooper (University of Queensland, Brisbane, AUS) and L. O'Neill (Trinity College, Dublin, IRL) and used at 10 μM, whereas the fatty-acid synthase (FASN) inhibitor C75 was bought from Sigma-Aldrich (C5490; Saint Louis, MO, USA) and used at 5 or 10 μM. The cathepsin L inhibitor was also purchased from Sigma-Aldrich (SCP0010) and used at 50 nM.

**Cell isolation and stimulation.** Peripheral blood mononuclear cells (PBMCs) from healthy volunteers and patients were isolated via centrifugation with Ficoll-Paque PLUS (GE Healthcare, Rockville, MD, USA). CD4$^+$ and CD8$^+$ T cells were subsequently isolated either via cell-sorting or via magnetic bead separation. For cell-sorting, PBMCs were treated for 10 min with human FcR blocking reagent (Miltenyi Biotec Ltd, Bisley, UK) and then stained with anti-human CD4, CD8a, and CD56. Cell-sorting was subsequently performed using a BD FACSAria™ II Cell Sorter (Flow Core facility, King's College London). CD4$^+$ and CD8$^+$ T lympho-cytes were sorted, respectively, as CD4$^+$CD56$^-$ and CD8$^+$CD56$^-$ cells with cell purity consistently ≥ 99%. For isolation via magnetic beads, contaminating CD56$^+$ natural killer and natural killer T cells were first depleted via positive selection using the CD56$^+$ Cell Isolation Kit (Miltenyi Biotec, Auburn, CA, USA), and CD8$^+$ or CD4$^+$ T cells were then purified using the CD8$^+$ or CD4$^+$ Cell Isolation Kit-II (both from Miltenyi Biotec, Auburn, CA, USA), following the manufacturer's instructions. The purity of final cell populations was consistently ≥ 95%.

For human T-cell stimulation, 48-well plates were coated overnight at 4 °C with 2 μg/mL of anti-CD3 (for CD4$^+$ T cells and 0.25 μg/mL for CD8$^+$ T cells), anti-

CD28 (2 μg/mL, BD Biosciences, Oxford, UK), and anti-CD46 (2 μg/mL GB24, generated in house) antibodies diluted in modified dPBS (Hyclone, Logan, UT, USA). T cells were resuspended in RPMI 1640 (Gibco Gaithersburg, MD, USA) supplemented with 1% penicillin-streptomycin, 200 mM L-glutamine (both from Sigma-Aldrich, Saint Louis, MO, USA), 10% fetal calf serum (Gibco, Gaithersburg, MD, USA), and 25 U/mL recombinant human IL-2 at $5 \times 10^5$ cells/well and stimulated for in an incubator at 37 °C and 5% $CO_2$ for time points indicated in the respective experiments. Note, were the usage of specific inhibitors or agonists were indicated in the experiments, cells were preincubated for 15 min with respective reagents or carrier control and then further cultured and activated.

**Cytokine measurements and detection of active caspase-1.** Cytokine secretion from activated human T cells into cell supernatants was quantified using the human Th1/Th2/Th17 Cytometric Bead Array (BD Biosciences, San Jose, CA, USA) following the manufacturer's instructions. Samples were acquired on a Fortessa LSRIII supported by FACS diva software (BD Biosciences, San Jose, CA, USA). Data were analyzed by FCAP array software (Soft Flow Hungary Ltd, Pecs, Hungary). IL-1β was measured using the Human IL1B Duo Set ELISA kit (R&D system Minneapolis, MN, USA) following the manufacturer's instructions in combination with the SIGMAFAST™ OPD tablets (Sigma-Aldrich, Saint Louis, MO, USA) for substrate detection. Generation of cleaved and active caspase-1 in human cells was monitored using the Green FLICA Caspase-1 Assay Kit (ImmunoChemistry Technologies, Bloomington, MN, USA) according to the manufacturer's protocol with subsequent FACS analysis.

**Cell proliferation assay.** For the measurement of CD8$^+$ T-cell proliferation, the CellTiter 96® AQ$_{ueous}$ One Solution Cell Proliferation Assay (MTS) from Promega (Madison, WI, USA) was used. This is a colorimetric method for determining the number of viable cells in proliferation, cytotoxicity, or chemosensitivity assays. Freshly purified CD8$^+$ T cells were either left non-activated or activated with immobilized activating antibodies to CD3 alone or to CD3 + CD28 or CD3 + CD46 in combination in 150 μL in 96-well plates. Cell proliferation was measured at days 1–5 by adding 20 microliter of the CellTiter 96® AQ$_{ueous}$ One Solution Reagent directly to culture wells, incubating for 1–4 h and then recording absor-bance at 490 nm with a 96-well plate reader. The quantity of formazan product as measured by the amount of 490 nm absorbance is directly proportional to the number of living cells in culture.

**Cytotoxicity assay.** The prostate cancer cell line DU145 (a gift from Dr. Hide Yamamoto, King's College London, UK) was used as source of target antigen

presenting cells to measure cytotoxic activity of human CD8$^+$ T cells. DU145 cells were cultured in RPMI 1640 media (Gibco, Gaithersburg, MD, USA), with 1% penicillin-streptomycin and 2 mM L-glutamine (all from Sigma-Aldrich, Saint Louis, MO, USA) in 24-well plates to 90% confluence. Bead-purified CD8$^+$CD56$^-$ T cells were labeled with 2.5 μM carboxyfluorescein succinimidyl ester (CFSE) (Molecular Probes, Eugene, OR, USA) in PBS and then either left non-activated or activated with antibodies to CD3 alone, CD3 + CD28 or CD3 + CD46 for 24 h and then plated onto the layer of DU145 cells at a 1 (DU145): 8 (T cell) ratio. After 12 h of co-culture, DU145 and T cells were harvested via trypsinization and cell mixtures stained with Annexin V in Annexin V binding buffer, followed by addition of 0.5 μg/mL propidium iodide (PI) (all from BD Biosciences, San Jose, CA, USA). Cells were analyzed using a FACS Calibur cytometer for Annexin V and PI-staining, supported by CellQuest Pro software (both from BD Biosciences, San Jose, CA, USA) with DU145 target cells identified as CFSE-negative cells. Annexin V and PI double-positive cells were considered in the evaluation of % of killed DU145 cells. Annexin V single-positive cells, Annexin V/PI double-positive cells and PI single-positive cells were identified as early apoptotic, late apoptotic, and necrotic cells, respectively.

**Flow cytometry**. Cells were stained for desired surface markers in PBS with 0.5% BSA and 2 mM EDTA (both from Sigma-Aldrich, Saint Louis, MO, USA), the addition of the appropriate antibody and isotype control antibody combinations as well as the LIVE/DEAD Cell Viability Assay (Life Technologies, Carlsbad, CA, USA) for 30 min on ice. For intracellular stainings, cells were first fixed and permeabilized utilizing the Cytofix/Cytoperm kit (BD Biosciences, San Jose, CA, USA) and preincubated for 10 min with human FcR blocking reagent (Miltenyi Biotec, Auburn, CA, USA) before addition of the desired antibodies and incubation for 30 min on ice. If the initial staining was performed with non-conjugated antibodies, cells were washed twice with PBS and then incubated for an additional 30 min on ice with a desired fluorochrome-conjugated secondary antibody or isotype control. Cells were washed again and data acquired utilizing a Fortessa LSRIII supported by FACS diva software (BD Biosciences, San Jose, CA, USA), and analyzed using the FlowJo 10.0.8 software (Ashland, OR, USA).

**Confocal microscopy**. T cells were fixed, permeabilized, and stained as described for intracellular stainings performed for flow analyses. Stained cells were mounted onto glass slides using VECTASHIELD media with DAPI (Vector labs, Burlingame, CA, USA), covered with a cover glass and sealed using nail polish. Images were subsequently acquired at room temperature with a Nikon A1R confocal microscope (Nikon Imaging Centre, KCL) using a 100 x oil immersion objective, and obtained pictures analyzed using the NIS Elements software version 4.03 (Nikon, Tokyo, JP).

**RNA extraction, cDNA synthesis, and RT-PCR**. RNA was extracted from T cells using the RNeasy Mini kit with on-column DNAse digestion (both from QIAGEN, Hilden, Germany), following the manufacturer's instructions. RNA concentration (A260) and purity (A260/A280 ≈ 2.0) was monitored using the spectrophotometer NanoDrop 1000 (Thermo Fisher, Waltham, MA). Reverse transcription to cDNA was performed utilizing the RevertAid H Minus First Strand cDNA Synthesis Kit according to the manufacturer's suggestions (Thermo Fisher Scientific, Waltham, MA, USA).

Real-Time PCR for *CD46 CYT-1* and *CYT-2*, *IL1B* and *ACTB* was performed using 1 x Taqman Universal Mastermix (Applied Biosystems, Foster City, CA, USA). *CYT-1* primers: forward 5′-TTGTCCCGTACAGATATCTTC–3′; reverse 5′-CTTCTCAGAGAGAAGTAAATTTT-3′; *CYT-1* antisense probe: 5′- AGTTAGGT ATGTGCCTTTCTTCTTCCTCC-3′; *CYT-2* primers: forward 5′- GGAGGAAGAA GAAAGGGAAAGC-3′; reverse 5′- TCAGCCTCTCTGCTCTGC-3′; *CYT-2* antisense probe: 5′- AGATGGTGGAGCTGAATATGCCACTT-3′; *IL1B* primers: forward 5′-CTCGCCAGTGAAATGATGGCT-3′; reverse: 5′-GTCGGAGATTC GTAGCTGGAT-3′; *ACTB* primers: forward: 5′-ACGGCCAGGTCATCACCATT G-3′; reverse: 5′-AGTTTCGTGGATGCCACAGGAC-3′. PCR amplification was performed in an ABI Prism 7900HT thermocycler (Applied Biosystems) for 2 min at 50 °C, 10 min at 95 °C, followed by 40 cycles of combined 15 s at 95 °C and 1 min at 60 °C. The sequence detector SDS 2.4 software (Applied Biosystems) was used to export the threshold cycles (Ct) values. Expression of the target gene was normalized using the endogenous gene 18S ribosomal RNA (18S rRNA). The relative quantification (RQ) of CYT-1 vs. CYT-2 presence was performed using the ΔΔCt method[47], where: ΔCT = Ct (CYT-1 or CYT-2) – Ct (18S rRNA), ΔΔCt = ΔCt (CD8 + cells) – (ΔCt CD4 + cells), RQ = $2^{-\Delta\Delta Ct}$.

Probes and primers used to detect *FASN, FABP5, SLC7A5*, and *RPL7* (for transcript expression normalization) transcripts were purchased from Thermo fisher. Their respective sequences are proprietary and are hence not publicly available. The hyperlinks below provide the padded amplicon sequence and denotes the general targeting area for each gene.

*FASN*
(Hs01005622_m1; http://www.ncbi.nlm.nih.gov/nuccore/NM_004104.4?report =fasta&log$ = seqview&format = text&from = 249&to = 549),
*FABP5*
(Hs02339439_g1;

http://www.ncbi.nlm.nih.gov/nuccore/NM_001444.2?report = fasta&log$ = seqview&format = text&from = 314&to = 614), and
*SLC7A5*
(Hs01001189_m1; http://www.ncbi.nlm.nih.gov/nuccore/NM_003486.5?report =&fasta&og$ = seqview&format = text&from = 1210&to = 1510)
*RPL7* (Hs02596927_g1; http://www.ncbi.nlm.nih.gov/nuccore/NM_000971.3? report = fasta&log$ = seqview&format = text&from = 165&to = 465).

The uncropped blot for the *IL1B* RNA measurement shown in Fig. 4e is displayed in Supplementary Fig. 8.

**CD46 RNA silencing**. siRNAs targeting *CD46* mRNA and control scrambled siRNA were purchased from Origene (Rockville, MD, USA) and delivered at a final concentration of 15 nM (with a mixture of three different siRNA at 5 nM each or scramble control at 15 nM) into primary human CD8$^+$ T cells by transfection with Lipofectamine RNAiMAX (Life Technologies, Paisley, UK) following the manufacturer's instructions in the presence of 50 ng/mL of recombinant human IL-15 and activating antibodies to CD3 (0.25 μg/mL) and CD28 (2 μg/mL) for 48 h.

**Microarray data generation and analysis**. Transcriptome profiling was performed by the KCL Genomic Centre (London, UK) utilizing Illumina chip Human HT-12 v4 Expression BeadChip (Illumina, San Diego, CA, USA) on RNA obtained from CD8$^+$ T cells freshly isolated via cell-sorting (CD8$^+$, CD4$^-$, CD56$^-$) from the blood of three different healthy donors. Purified T cells were activated with immobilized antibodies to CD3 (0.25 μg/mL) either alone or in combination with immobilized antibodies to CD46 (2 μg/mL) for 6 h in media supplemented with 25 U/mL of recombinant human IL-2. Expression data were analyzed with the Partek Genomics Suite (Partek Inc., St. Louis, MO, USA) version 6.6 and Gene Set Enrichment Analysis, GSEA (Broad Institute of MIT and Harvard, USA) with a normalized enrichment score of 1.8 to derive normalized enrichment score (NES), nominal *p*-value and FDR *q*-value. Supplementary Excel File 3 summarizes the list of annotated genes and normalized read values from microarrays and volcano plot for Supplementary Fig. 6a, b.

**RNA-Seq data generation and analysis**. The sequencing libraries were constructed from 10 pg–10 ng of total RNA using the Takara Bio USA's SMART-Seq V4 Ultra Low Input RNA Kit for Sequencing (Cat # 634888) following the manufacturer instruction. Briefly, full-length cDNA synthesis was performed using oligo-dT and SMART oligos and the Illumina sequencing adaptors and barcodes incorporated into the final library by PCR. The fragment size of the RNAseq libraries was verified using the Agilent 2100 Bioanalyzer (Agilent, Santa Clara, CA, USA) and the concentrations determined using the Qubit instrument (Life Technologies, Carlsbad, CA, USA). The libraries were loaded onto the Illumina HiSeq 3000 machine (Illumina, San Diego, CA, USA) for 1 × 50 bp single end read sequencing and generated about 55 M reads per sample. The. fastq files were generated using the bcl2fastq software (Illumina) for further analysis. Quality controls of raw sequencing reads were assessed using FastQC tools[48]. If required, adapter sequences and low-quality bases were trimmed using Cutadapt[49]. Reads were mapped against the reference human genome (GRCh38) [50] using STAR[51] and read counts for each gene in GENECODE annotation v25 were generated using featureCounts[52]. Differential expression analysis was performed using the DESeq2 R package[53]. Differentially expressed genes were considered as those with fold change of at least 2 in either direction at *q*-value of 0.05 or less. Gene set enrichment tests for canonical pathways were performed using GSEA (Broad institute)[54].

**Statistical analysis**. Statistical analyses were performed using regression analysis, the One-way analysis of variance (ANOVA) with Tukey Multiple Comparison test or the Paired/Unpaired Student's *t*-test and Bonferroni correction where appropriate (GraphPad Prism software, La Jolla, CA, USA). Data were assessed for normal distribution using the D'Agostino-Pearson test. Data were expressed as mean ± standard error of the mean (SEM), and statistical significance was attributed to *p* values < 0.05.

## Data availability

The sequencing and microarray data have been deposited with the Gene Expression Omnibus under the Accession number GSE119919.

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

## Acknowledgements

We thank the patients and the healthy donors for their support. We also thank the Bioinformatics and Computational Biology Core as well as the DNA Sequencing and Genomics Core at NHLBI for sample processing and help with the analyses. This work was financed by the MRC Centre grant MR/J006742/1, an EU-funded Innovative Medicines Initiative BTCURE (C.K.), a Wellcome Trust Investigator Award (C.K), a Wellcome Trust Clinical Fellowship (B.A.), the King's Bioscience Institute at King's College London (G.A.), a grant from the Guy's and St. Thomas' Charity (C.K., B.A., and N.P.), The King's College London BRC Genomics Facility, the National Institute for Health Research (NIHR) Bio-medical Research Centre based at Guy's and St. Thomas' NHS Foundation Trust and King's College London, [in part] by the Intramural Research Program of the NIH, the National Institute of Diabetes and Digestive and Kidney Diseases (NIDDK), and by the Division of Intramural Research, National Heart, Lung, and Blood Institute, NIH.

## Author contributions

H.L., G.A. and C.K. conceived and directed the study, performed experiments, and wrote the manuscript. G.L.F. designed, performed and analyzed the CD46 CYT-1 and CYT-2 expression studies. E.W., J.R., and B.A. designed, performed, and analyzed CD8+ T-cell metabolic inhibition experiments and—in conjunction with A.S., L.C., V.F.-B., A.F., P.R. W., D.K., P.P., L.P., and N.P.—functional experiments involving samples from the CD46-deficient patients and matched healthy donors. Y.L., P.L., M.P., and I.T. performed and analyzed the RNA-Seq experiments. N.N. and J.R. designed, performed, and analyzed all Q-PCR experiments, and B.A. and P.L. designed, performed, and analyzed the gene array

experiments. All authors discussed and edited the manuscript. H.L. and C.K. contributed equally to the work and are shared senior authors.

## Additional information

**Competing interests:** The authors declare no competing interests.

