## [Peer Review File · Nature Communications]

Reviewers' comments:

Reviewer #1 (TCR signaling, co-stimulation)(Remarks to the Author):

The paper by Arbore and co-authors determined the role of the complement activation and inflammasome crosstalk on the effect of cytotoxic CD8+ T lymphocytes. The authors demonstrate that the complement receptor CD46 acts as an essential co-stimulatory molecule for CTL activity, while NLRP3 is not required for this process. Thus, this paper gives a novel view on CD8+ T cell activity. It has recently been shown that next to the CD28/B7 ligation additional signals are required for induction of Th1 cells. Furthermore, autocrine CD46 engagement during T cell stimulation affects metabolic pathways, which are important for interferon- γ secretion, while overreactive Th1 responses could be corrected by a NLRP3 inhibitor. This manuscript extends these studies on the role of complement from CD4+ to CD8+ T cell responses suggesting that autocrine complement and the inflammasomes play distinct roles in human T cell subpopulations. This is an interesting topic and the authors performed their experiments very carefully with all required controls. There are some issues, which have to be addressed by the authors:

- It would have been helpful to have at least two more patients with CD46 deficiencies. This is important for the impact of the respective results obtained.
- Gene array analysis should have been performed with CD46 deficient patient material to better understand the role of CD46 in this context.
- Expression of the differentially expressed genes identified by array analysis of CD3, CD3+ CD46 activated CD8+ T cells should be tested in patients' material.

Reviewer #2 (CD4/CD8 function, T cell signalling)(Remarks to the Author):

The authors examine the role of the complement receptor CD46 in human CD8+ T cell co-stimulation. Building on their previous evidence that CD46 is a highly potent and necessary co-stimulus for human CD4+ T cells, and their data on the pathways induced following CD46 engagement, these studies are of considerable interest. Surprisingly, while CD46 can serve as a strong co-stimulator for cytokine production and cytotoxic activity of human CD8+ T cells, analysis of CD46-deficient patients and CD46 gene silencing indicated CD46 expression is not necessary for these responses. This differs from CD4+ T cells, which showed greatly depressed Th1 cytokine production in the absence of CD46, in response to various stimuli. Mechanistically, these findings correlate with a lack of conventional NLRP3 activation following CD46 engagement in CD8+ T cells, while this pathway was induced in CD4+ T cells, in which it was important for cytokine production (confirming this groups' previous reports). Likewise, CAPS patients with gain-of-function mutations in NLRP3 showed no substantial dysregulation of CD8+ T cell responses. Rather, the authors ascribe the primary beneficial role of CD46 in CD8+ T cells to the induction of metabolic pathway, including fatty acid synthesis.

These studies are well conducted and utilize both analysis of patients with genetic mutations in key pathways as well as genetic and biochemical manipulation of this pathway in normal human CD4+ and CD8+ T cells. The authors anticipate likely concerns (such as the frequency of naive vs memory CD8+ T cells in CD46-deficient humans) and their data are compelling. There are some remaining concerns, however.

1) The studies in Fig. 5 are used to argue that the chief impact of CD46 costimulation in CD8+ T cells is to promote metabolic pathways, including fatty acid synthesis. While it is clear that the blockade of FASN (Fig. 5e) does indeed impair IFN γ and degranulation of CD8+ T cells costimulated with CD46,

there is also highly potent (and significant) blockade of cells stimulated with CD3+CD28 or CD3 alone. There doesn't, then, seem to be a qualitative change in the impact of this pathway when CD46 costimulation is involved, which makes it hard to judge whether induction of this pathway should be regarded as a unique characteristic of cells activated with CD46 costimulation.

2) The previous point raises a more general question of how we should compare these different stimulation conditions – is CD46 a more potent costimulatory regardless of stimulation dose in CD8+ T cells? Or would increased CD3 or CD3+CD28 stimulation (e.g. using higher doses of the antibodies) yield qualitatively and quantitatively similar outcomes as stimulation with CD3+CD46? These concerns were minimized in the authors previous and current data on CD4+ T cells, where CD46 deficiency or knockdown had such substantial effects on responses to other stimuli (e.g. impaired CD3+CD28 responses). But in the case of CD8+ T cells, it is unclear to this reviewer that there are demonstrable qualitative differences in CD46 and CD28 co-stimulation. To some extent, one could consider that the lack of any discernable changes in the CD8+ T cell populations of CD46-deficient patients (e.g. Supp. Fig. 2) supports the idea that, in vivo, the lack of CD46 costimulation has minimal impact on the CD8+ T cell pool (while clearly having substantial effects on the CD4+ T cell pool). This is not to take from the impact of this report, but may lead to the conclusion that there is even less of a non-redundant role of CD46 in CD8+ T cells than proposed.

3) Whilst this reviewer appreciates the limitations in obtaining samples from CD46-deficient patients, the lack of statistical analysis in the data from Fig. 2b makes interpretation difficult. These data, from duplicate samples, suggest a trend by which lack of CD46 impairs the IFN γ production by CD8+ T cells in response to CD3+CD28 stimulation, and it would be valuable to know whether this can be confirmed with further repeats. This relates to the previous points – in that clear demonstration of depressed CD3+CD28 responses by CD46-deficient CD8+ T cells (assuming no changes in TCR or CD28 expression levels) would swing the argument back to saying that CD46 does have a key but distinct role in both CD4+ and CD8+ T cells.

We appreciate the time the reviewers have taken to review our manuscript and for providing invaluable input to help us improve the content. We are also pleased that the reviewers overall were positive about the quality and scope of our study.

The two key issues that needed to be addressed based on the reviewers' comments were A. To assess the CD8⁺ T cells response from a minimum of two additional patients with CD46 deficiency (including functional and gene expression studies to confirm the importance of CD46 in driving cell metabolic pathways during cell activation), and B. To define whether there are qualitative or only quantitative differences between CD28 and CD46 co-stimulation during human CD8⁺ T cell activation.

To A.: We are pleased to report that we have been able to obtain blood samples from three additional patients with CD46 deficiency (total of $n = 4$). The data obtained utilizing the CD8⁺ T cells from these patients align fully with those from the initial patient assessed and together now strongly support the conclusion that CTL activity is significantly reduced when CD46 is inadequately expressed (Figure 3e,f of the revised manuscript).

To B.: We had previously demonstrated that, in human CD4⁺ T cells, CD28 co-stimulation potentiates TCR-induced cell-intrinsic generation of the CD46 ligand C3b, which signals in an autocrine loop to engage CD46 and induce Th1 differentiation (Kolev *et al.*, *Immunity*, 2015). We have generated here now additional data suggesting that an equivalent 'autocrine CD46 activation pathway' also exists in human CD8⁺ T cells: CD8⁺ T cells contain intracellular stores of the C3aR, C3, and C3a and activation of CTLs with increasing amounts of anti-CD3 and/or anti-CD28 antibodies is associated with greater surface deposition of C3b together with simultaneous and proportional increments in cytotoxic activity and IFN- γ production (new Figure 2c,d). Aligning with these data, activation of CTLs in the presence of a cathepsin L inhibitor (which prevents intracellular C3 activation) significantly reduced CTL C3b surface levels, degranulation and IFN- γ (new Supplementary Figure 2). Importantly, CD8⁺ T cells from all four CD46-deficient patients assessed here express normal levels of CD3 and CD28 but have significantly depressed CTL responses upon both CD3+CD28 and CD3+CD46 activation. Together, these findings suggest that one of the qualitatively distinct roles of CD28 in human CD8⁺ T cells (and similar to its role in CD4⁺ T cells) is to function upstream of CD46 as modulator of autocrine CD46-dependent signaling events, which in turn supports metabolic reprogramming required for optimal effector function.

Of note, we have added further depth to our evaluation of CD46's role on CTL activity by now also including an analysis of separated naive and memory CD8⁺ T cells demonstrating that CD46 co-stimulation has comparable effects on these CTL subpopulations (Figure 1f).

We have marked changes in the text body in the revised manuscript in 'red'.

Our additional, point-by-point, replies to the reviewers are as follows:

Reviewer #1:

The paper by Arbore and co-authors determined the role of the complement activation and inflammasome crosstalk on the effect of cytotoxic CD8⁺ T lymphocytes. The authors demonstrate that the complement receptor CD46 acts as an essential co-stimulatory molecule for CTL activity, while NLRP3 is not required for this process. Thus, this paper gives a novel view on CD8⁺ T cell activity. It has recently been shown that next to the CD28/B7 ligation additional signals are required for induction of Th1 cells. Furthermore, autocrine CD46 engagement during T cell stimulation affects metabolic pathways, which are important for interferon- γ secretion, while overreactive Th1 responses could be corrected by an NLRP3 inhibitor. This manuscript extends these studies on the role of complement from CD4⁺ to CD8⁺ T cell responses suggesting that autocrine complement and the inflammasomes play distinct roles in human T cell subpopulations. This is an interesting topic and the authors performed

their experiments very carefully with all required controls. There are some issues, which have to be addressed by the authors:

1. It would have been helpful to have at least two more patients with CD46 deficiencies. This is important for the impact of the respective results obtained.

We fully agree with the reviewer's suggestion that the analysis of blood samples from at least two more patients with CD46 deficiency - which was also requested by reviewer #2 – would strengthen the conclusion and impact of this study.

We are pleased to report that we were able to obtain samples from three additional patients with CD46 deficiency. Importantly, the functional analyses (as well as RNA-Seq analyses, please see below) of CD8⁺ T cells isolated from these additional patients fully confirmed the observations we had made using T cells from our 'initial' patient: all parameters of 'healthy' CTL activities, including IFN- γ secretion, degranulation and killing are significantly impaired in the patients' cells (Figure 3e and f; Supplementary Figure 3a – d).

Furthermore, we have extended our work and included new data demonstrating that CD46 stimulation is superior to CD28 co-stimulation in both naïve and memory human CD8⁺ T cells (Figure 1f).

We believe that, together, these data strongly support the interpretation that CD46 is a key co-stimulator for optimal human CTL activity.

2. Gene array analysis should have been performed with CD46 deficient patient material to better understand the role of CD46 in this context.

As suggested by the reviewer, we have performed additional gene expression analysis experiments (RNA-Seq) using purified CD8⁺ T cells from three CD46-deficient patients (CD46-2 to -4) to better understand the impact of CD46 activation on CTLs.

CD46-deficient patients are rare and the obtained blood sample sizes yielded a limited number of pure CD8⁺ T cells. As a substantial proportion of these cells were required to analyze the functional phenotype of the patients' CTLs, we were unfortunately not able to assess gene expression for all four 'ideal' activation conditions (non-activated, CD3, CD3+CD28, and CD3+CD46) but only for non-activated and CD3+CD46-activated CTLs for the RNA-Seq analyses. We, however, argue that – since the patients lack detectable CD46 but have normal TCR expression – their CTL activation profile should reflect TCR stimulation 'only' gene induction.

Indeed, these new data confirmed our initial experiments in which we compared CD3 vs. CD3+CD46-activated CTLs from healthy donors (now Supplementary Figure 6a and b): lack of CD46 leads to 1. significantly reduced gene expression in TCR-stimulated CTLs and, 2. the pathways affected fall mostly into those of metabolic nature including lipid metabolism (FASN and FABP5) and transmembrane transport of small molecules (nutrient influx, SLC7A5) (Figure 6a – c; Supplementary Excel Files 1 – 3). Thus, these new analyses further support our notion that CD46 co-stimulation in human CTLs functions via augmentation of nutrient influx and fatty acid metabolism. However, our data also indicate that there are clearly additional CD46-driven pathways 'at work' in stimulated CTLs (Supplementary Excel Files 1 – 3) and it will be exciting to further dissect those in the future to more fully understand CD46's mechanistic contributions to normal CTL biology.

3. Expression of the differentially expressed genes identified by array analysis of CD3, CD3+ CD46 activated CD8⁺ T cells should be tested in patients' material.

We have now performed the requested experiments and found that Q-PCR analyses showed indeed a significant diminution of FASN and FABP5 induction – and a trend for reduced SLC7A5 expression - in CD3+CD46-stimulated CTLs from CD46-deficient patients when compared to healthy donors. These new data are now included into Figure 6g.

Reviewer #2:

The authors examine the role of the complement receptor CD46 in human CD8+ T cell co-stimulation. Building on their previous evidence that CD46 is a highly potent and necessary co-stimulus for human CD4+ T cells, and their data on the pathways induced following CD46 engagement, these studies are of considerable interest. Surprisingly, while CD46 can serve as a strong co-stimulator for cytokine production and cytotoxic activity of human CD8+ T cells, analysis of CD46-deficient patients and CD46 gene silencing indicated CD46 expression is not necessary for these responses. This differs from CD4+ T cells, which showed greatly depressed Th1 cytokine production in the absence of CD46, in response to various stimuli. Mechanistically, these findings correlate with a lack of conventional NLRP3 activation following CD46 engagement in CD8+ T cells, while this pathway was induced in CD4+ T cells, in which it was important for cytokine production (confirming this groups' previous reports). Likewise, CAPS patients with gain-of-function mutations in NLRP3 showed no substantial dysregulation of CD8+ T cell responses. Rather, the authors ascribe the primary beneficial role of CD46 in CD8+ T cells to the induction of metabolic pathway, including fatty acid synthesis.

These studies are well conducted and utilize both analysis of patients with genetic mutations in key pathways as well as genetic and biochemical manipulation of this pathway in normal human CD4+ and CD8+ T cells. The authors anticipate likely concerns (such as the frequency of naïve vs memory CD8+ T cells in CD46-deficient humans) and their data are compelling. There are some remaining concerns, however.

1) The studies in Fig. 5 are used to argue that the chief impact of CD46 costimulation in CD8+ T cells is to promote metabolic pathways, including fatty acid synthesis. While it is clear that the blockade of FASN (Fig. 5e) does indeed impair IFN γ and degranulation of CD8+ T cells costimulated with CD46, there is also highly potent (and significant) blockade of cells stimulated with CD3+CD28 or CD3 alone. There doesn't, then, seem to be a qualitative change in the impact of this pathway when CD46 costimulation is involved, which makes it hard to judge whether induction of this pathway should be regarded as a unique characteristic of cells activated with CD46 costimulation.

We thank the reviewer for her/his insightful comments.

As mentioned above in our answer to Point A., we have now generated new data showing that, similar to what we have previously shown in human CD4+ T cells (Kolev *et al.*, Immunity, 2015), the strength of TCR engagement and CD28 co-stimulation directs the level of intrinsic C3b generation and autocrine CD46 engagement with subsequent dose-dependent CD46-driven augmentation of CTL activation (new Figure 2). Furthermore, as in CD4+ T cells, the inhibition of cathepsin L activity (which cleaves C3 into C3a and C3b (Liszewski *et al.*, Immunity, 2015)) in activated CD8+ T cells leads to a proportional reduction of C3b generation and CTL activity (new Supplementary Figure 2).

Thus, one of the specific roles of CD28 in CD4+ and CD8+ T cells is to generate the ligand for autocrine CD46 activation and via this to then induce specific CD46-mediated downstream (metabolic) signals that support respective T cell effector functions (we have also updated our results, discussion and model schematic in Figure 6h to reflect these new insights). Under this scenario, it is then fully expected (and we have indeed shown this for CD46-driven mTORC1 activation in CD4+ T cells (Kolev *et al.*, Immunity, 2015)) that inhibition of FASN also affects the CD3 and CD3+CD28 activated CTLs as FASN activity under those latter conditions is driven by CD28-mediated autocrine C3b/CD46 stimulation.

2) The previous point raises a more general question of how we should compare these different stimulation conditions – is CD46 a more potent costimulatory regardless of stimulation dose in CD8+ T cells? Or would increased CD3 or CD3+CD28 stimulation (e.g. using higher doses of the antibodies) yield qualitatively and quantitatively similar outcomes as stimulation with CD3+CD46? These concerns were minimized in the authors previous and current data on CD4+ T cells, where CD46 deficiency or knockdown had such substantial effects on responses to other stimuli (e.g. impaired CD3+CD28

responses). But in the case of CD8⁺ T cells, it is unclear to this reviewer that there are demonstrable qualitative differences in CD46 and CD28 co-stimulation. To some extent, one could consider that the lack of any discernable changes in the CD8⁺ T cell populations of CD46-deficient patients (e.g. Supp. Fig. 2) supports the idea that, *in vivo*, the lack of CD46 costimulation has minimal impact on the CD8⁺ T cell pool (while clearly having substantial effects on the CD4⁺ T cell pool). This is not to take from the impact of this report, but may lead to the conclusion that there is even less of a non-redundant role of CD46 in CD8⁺ T cells than proposed.

Please, see our 'combined' response to Points 2 and 3 below under Point 3.

3) Whilst this reviewer appreciates the limitations in obtaining samples from CD46-deficient patients, the lack of statistical analysis in the data from Fig. 2b makes interpretation difficult. These data, from duplicate samples, suggest a trend by which lack of CD46 impairs the IFN γ production by CD8⁺ T cells in response to CD3+CD28 stimulation, and it would be valuable to know whether this can be confirmed with further repeats. This relates to the previous points – in that clear demonstration of depressed CD3+CD28 responses by CD46-deficient CD8⁺ T cells (assuming no changes in TCR or CD28 expression levels) would swing the argument back to saying that CD46 does have a key but distinct role in both CD4⁺ and CD8⁺ T cells.

As suggested by the reviewer, we have now titrated the TCR and CD28 versus CD46 signal strength in our *in vitro* system and observed that CD46 remained the most potent co-stimulator under all conditions tested (new Figure 2). This is likely due to the fact that direct engagement of CD46 by a specific antibody – as opposed to TCR and CD28-driven C3b generation and autocrine CD46 ligation – leads to more potent cross-linking and signaling of CD46. However, the fact that the patients' CTL responses are depressed even under CD3+CD28 activation conditions (Figure 3) indicates strongly that CD46 signals are distinct from CD28 signals and that they are indeed a requirement also for 'optimal' CTL activity.

We agree with the reviewer that our data overall confirm that CD46-dependent signals play both key and distinct roles in the biology of CD4⁺ versus CD8⁺ T cells. This is, indeed, substantiated by the limited overlap in genes induced by this pathway in the two populations, despite broadly regulating similar pathways such as cellular metabolism. It is tempting to speculate that this may reflect inherent differences between CTLs, that have a more limited functional repertoire vs. T helper cells, which undergo lineage decisions and have a more broad range of capabilities. During evolution, CD46 may have simply assumed a more prominent role in the complex networks regulating CD4⁺ T cell activities.

The reviewer also noted correctly that the CD46-deficient patients lack discernable changes in circulating CTL population markers we so far analyzed *ex vivo*. In view of the recurrent viral infections in these patients, we were also surprised by this finding. The future direction of our research aims to determine whether CD46 regulates the development or persistence of (tissue-resident) memory cells, cellular senescence/apoptosis and how CD46 regulates different target genes in CD4⁺ vs. CD8⁺ T cells. We hope that these investigations will address the 'big picture' question raised by the reviewer.

REVIEWERS' COMMENTS:

Reviewer #1 (Remarks to the Author):

The revised version of the paper by Arbore and co-authors improved a lot.

- The authors included novel results, such as CD46 induced IFN- γ secretion, CD107A and granzyme B positive cells.
- In addition, they included analyses of three CD46 deficient patients, which confirmed their results concerning reduced IFN- γ production upon co-stimulation and lack of CD46-mediated increase in degranulation and granzyme B expression.
- They included cytotoxicity assays.
- Furthermore, gene expression profiles of CD3+ CD46 activated T cells from three CD46 deficient patients and three healthy controls were performed. These data demonstrate that CD46 induced in particular genes associated with metabolic pathways.

Following minor concerns should be discussed:

- Why could the induced changes of FASN, FABP5 and SLC7A5 transcription alteration could not be identified by RNAseq.
- They extended their analyses to SLC7A5 coding for LAT1, which should be discussed.

Reviewer #2 (Remarks to the Author):

The authors have responded appropriately to previous concerns, adding new data and revising the text such that manuscript is much improved and the basis for the authors' hypothesis strengthened.

Reviewer #1:

The revised version of the paper by Arbore and co-authors improved a lot. The authors included novel results, such as CD46 induced IFN- γ secretion, CD107A and granzyme B positive cells. In addition, they included analyses of three CD46 deficient patients, which confirmed their results concerning reduced IFN- γ production upon co-stimulation and lack of CD46-mediated increase in degranulation and granzyme B expression. They included cytotoxicity assays. Furthermore, gene expression profiles of CD3+ CD46 activated T cells from three CD46 deficient patients and three healthy controls were performed. These data demonstrate that CD46 induced in particular genes associated with metabolic pathways.

We thank the reviewer for the overall positive evaluation of the additional data generated during the revision process of the manuscript.

Following minor concerns should be discussed:

1. Why could the induced changes of FASN, FABP5 and SLC7A5 transcription alteration could not be identified by RNAseq.

We may have not stressed this enough but the failure of CD46-deficient CTLs to induce normal levels of FASN, FABP5, and SCL7A5 has indeed been identified by our additional RNA-Seq. Please see our new Figure 6c (the three genes in question have been labeled in red letters). We further included a confirmation by real time PCR demonstrating that the clear reduction of the expression of these three genes in the patients' T cells in the new Figure 6g.

2. They extended their analyses to SLC7A5 coding for LAT1, which should be discussed.

The analysis of the LAT1 expression as well as a detailed rationale as to why we focused on this channel was already included in the original submission (please see pages 9 and 10 of the text in our original submission) of the manuscript and is not an extension that we included into the revised version of the manuscript.

Reviewer #2:

The authors have responded appropriately to previous concerns, adding new data and revising the text such that manuscript is much improved and the basis for the authors' hypothesis strengthened.

We are pleased that we have been able to address the reviewer's previous concerns in full and thank her/him again for the constructive suggestions.